# RSL24D1 sustains steady-state ribosome biogenesis and pluripotency translational programs in embryonic stem cells

Sébastien Durand[1,2,3,12], Marion Bruelle[1,12], Fleur Bourdelais[1,2,3,4,12], Bigitha Bennychen[5,6], Juliana Blin-Gonthier[7], Caroline Isaac[1,2,3], Aurélia Huyghe[1,2,3,8], Sylvie Martel[1,2], Antoine Seyve[1,2,9], Christophe Vanbelle[1,2], Annie Adrait[10], Yohann Couté[10], David Meyronet[1,2,11], Frédéric Catez[1,2,3], Jean-Jacques Diaz[1,2,3], Fabrice Lavial[1,2,3,8], Emiliano P. Ricci[7], François Ducray[1,2,9] & Mathieu Gabut[1,2] ✉

Embryonic stem cell (ESC) fate decisions are regulated by a complex circuitry that coordinates gene expression at multiple levels from chromatin to mRNA processing. Recently, ribosome biogenesis and translation have emerged as key pathways that efficiently control stem cell homeostasis, yet the underlying molecular mechanisms remain largely unknown. Here, we identified RSL24D1 as highly expressed in both mouse and human pluripotent stem cells. RSL24D1 is associated with nuclear pre-ribosomes and is required for the biogenesis of 60S subunits in mouse ESCs. Interestingly, RSL24D1 depletion significantly impairs global translation, particularly of key pluripotency factors and of components from the Polycomb Repressive Complex 2 (PRC2). While having a moderate impact on differentiation, RSL24D1 depletion significantly alters ESC self-renewal and lineage commitment choices. Altogether, these results demonstrate that RSL24D1-dependant ribosome biogenesis is both required to sustain the expression of pluripotent transcriptional programs and to silence PRC2-regulated developmental programs, which concertedly dictate ESC homeostasis.

Pluripotent stem cells (PSCs), including embryonic stem cells (ESCs) and induced pluripotent stem cells (iPSCs), have the unique abilities to self-renew in a naive state while remaining competent to differentiate into all lineages of developing embryos. This ambivalent state is tightly coordinated at different steps of gene expression, which have been extensively described at the chromatin, transcriptional and post-transcriptional levels[1–4]. This led to the identification of key regulatory epigenetic and transcriptional programs that rapidly rewire gene expression and regulate cell fate transitions in response to environmental cues[2,5,6]. For instance, chromatin modifications mediated by the

---

[1]Cancer Initiation and Tumoral Cell Identity (CITI) Department. Cancer Research Centre of Lyon (CRCL) INSERM 1052, CNRS 5286, Université Claude Bernard Lyon I, Centre Léon Bérard, Lyon, France. [2]Institut Convergence Plascan, Lyon, France. [3]Labex Dev2Can, Lyon, France. [4]Inovarion, 75005 Paris, France. [5]Dept. of Biochemistry, Microbiology and Immunology, Faculty of Medicine, University of Ottawa, Ottawa, ON K1H 8M5, Canada. [6]Human Health Therapeutics Research Centre, National Research Council Canada, Ottawa, ON K1A 0R6, Canada. [7]Laboratoire de Biologie et de Modélisation de la Cellule, ENS de Lyon, CNRS UMR 5239, Inserm U1293, Lyon, France. [8]Equipe labellisée la Ligue contre le cancer, Lyon, France. [9]Neuro-oncology department, Hospices Civils de Lyon, Lyon, France. [10]University Grenoble Alpes, INSERM, CEA, UA13 BGE, CNRS, CEA, FR2048, 38000, Grenoble, France. [11]Institut de Pathologie Est, Hospices Civils de Lyon, Lyon, France. [12]These authors contributed equally: Sébastien Durand, Marion Bruelle, Fleur Bourdelais. ✉e-mail: mathieu.gabut@inserm.fr

Polycomb Repressive Complexes 1 and 2 (PRC1 and PRC2) play pivotal roles in the transcriptional regulation of pluripotency and differentiation of PSCs[2,5–8]. However, multiple studies suggest that messenger RNA (mRNA) and protein levels are poorly correlated, including in PSCs, thus highlighting the importance of translation regulations for shaping the cellular proteome landscape required for cell fate changes[9–13]. In addition, recent studies highlighted a lower translation efficiency in undifferentiated ESCs, but also in adult stem cells, compared to differentiated progenies[14]. Therefore, this compelling evidence establish protein synthesis regulation as a key player in defining stem cell fate.

In eukaryotes, ribosome biogenesis is a complex and multistep process that involves over 280 ribosome biogenesis factors (RBFs) and different families of non-coding RNAs, consecutively in the nucleoli, the nucleoplasm, and the cytoplasm[15,16]. Many studies have established that the ribosome biogenesis is finely regulated in stem cells and may directly control stem cell properties[14]. Despite a low protein synthesis activity, ESCs display higher levels of rRNA transcription compared to endoderm-committed cells[17], and rRNA transcription is finely coordinated with proliferation rates and protein synthesis in mouse ESCs[18,19].

Similarly, several key RBFs are expressed at higher levels in ESCs compared to differentiated progenies and promote ESC self-renewal[20–22]. In addition to these essential RBFs, ribosome subunit-specific RBFs are also required to support ESC maintenance. Indeed, several factors implicated in the maturation of the 40S small ribosome subunit (SSU) are preferentially expressed in naive ESCs compared to differentiated progenies and support the translation of key pluripotency transcription factors (PTFs) such as NANOG[23]. Notably, NOTCHLESS, a RBF of the 60S large ribosome subunit (LSU), is required for the inner cell mass survival during early mouse embryogenesis. However, in contrast to NOTCHLESS functions in adult stem cells homeostasis, it is unclear whether its role in ribosome biogenesis is implicated in this developmental context[24–26]. Therefore, to date, the contributions of ribosome subunit-specific RBFs, especially pre-60S RBFs, to the steady-state stoichiometry of the 40S and 60S subunits and their impact on the regulation of PSC fate decision remain elusive and should be further investigated.

Here, we show that RSL24D1, a conserved homolog of the yeast pre-60S maturation and export factor Rlp24, is expressed at high levels in mouse and human PSCs compared to differentiated progenies and is essential for the maturation of the LSU in pluripotent mouse ESCs. RSL24D1 is also required to maintain a steady-state level of translation, particularly of key PTFs such as NANOG and POU5F1, but also of PRC2 factors that maintain repressive H3K27me3 marks over developmental genes. Moreover, high levels of RSL24D1 are required to support mouse ESC proliferation and self-renewal, while its depletion only has a moderate impact on lineage commitment and ESC differentiation. Altogether, these results establish that a bona fide 60S biogenesis and the resulting translation activity are coordinated with transcription and chromatin regulation networks to control ESC homeostasis.

## Results

### Rsl24d1 expression is enriched in murine and human PSCs
To identify factors contributing to ribosome assembly in PSCs, we first determined the mRNA expression profiles of 303 genes, including RBFs and ribosomal proteins (RPs), in murine PSCs and differentiated cell lines using publicly available RNA-seq data[27] (Supplemental Fig. S1a and Supplementary Data 1). Although the majority (70%) of these factors were expressed at higher RNA levels (fold change >1.5) in pluripotent cells compared to differentiated cells, Rsl24d1, a predicted ribosome biogenesis factor, displayed the most striking enrichment in PSCs (fold change >10[4]) (Fig. 1a).

We next confirmed that, similar to POU5F1, the RSL24D1 protein was also expressed at a higher level in mouse pluripotent CGR8 ESCs cultured either in serum+LIF (ESC[FBS]) or in 2i-induced naïve ground state (ESC[2i]) conditions compared to in vitro ESC-derived 12-days old differentiated embryoid bodies (EB[12]) (Fig. 1b). In contrast, the expression of RPL8, a canonical RP of the LSU, remained globally unchanged at the protein level as ESCs differentiate (Fig. 1b). More precisely, the dynamics of expression of Rsl24d1 mRNA is correlated to a rapid downregulation of several key PTFs, including Pou5f1 and Nanog, during the kinetics of ESC differentiation (Supplemental Fig. S1b). Similarly, RSL24D1 protein was consistently expressed at higher levels in ESC[FBS] conditions and progressively decreased after differentiation initiation to reach the lowest expression in EB[12] (Supplemental Fig. S1c). The expression of additional RPs also decreased, yet to a lower extent than RSL24D1, while the downregulation of the RBFs EIF6 and GTPBP4 (NOG1) rather followed RSL24D1's profile.

In addition to CGR8, we assessed the expression of RSL24D1 in two independent mouse ESCs lines (R1 and G4) cultured in similar conditions (ESC[2i], ESC[FBS], EB[12]) (Fig. 1c). Although RSL24D1 was expressed at different basal levels in the three ESC models, these results confirmed that RSL24D1 levels were significantly higher in pluripotent ESCs than in ESC-derived differentiated EBs. Altogether, these results firmly demonstrate that RSL24D1 expression is high in murine ESCs and strongly decreases upon differentiation.

We then hypothesized that the expression of Rsl24d1 would rather be determined by the pluripotency status than by the embryonic origin. To evaluate this hypothesis, iPSCs were obtained after ectopic expression of POU5F1, KLF4, c-MYC, and SOX2 in mouse embryonic fibroblasts (MEFs)[28] and the kinetics of somatic reprogramming was confirmed by the activation of endogenous Pou5f1 and Nanog mRNAs (Supplemental Fig. S1d). While RPL8 expression remained globally unchanged, RSL24D1 expression was highly increased in 14-day old iPSCs compared to parental MEFs (Fig. 1d). Since the early steps of iPSC formation are stochastic[29], we next investigated Rsl24d1 mRNA expression in cells with enhanced reprogramming potential at the single cell level from published data[30]. Interestingly, compared to cells engaged in early steps of the reprogramming path (eRP), Rsl24d1 expression was strongly enhanced in a continuum of cells representing different stages of active reprogramming (pre-PCs) and its levels were even further increased in chimera-competent reprogrammed cells (PCs) expressing high levels of PTFs (Supplemental Fig. S1e). Altogether, these observations suggest that Rsl24d1 expression is significantly enriched in mouse PSCs regardless of their embryonic origins.

We next investigated whether Rsl24d1 expression pattern was evolutionarily conserved in human PSCs. RNA-seq data first indicated that, similar to their murine counterparts, human PSCs expressed higher levels of RSL24D1 mRNAs compared to differentiated cell lines or tissues (Fig. 1e)[27]. Western blot analyses of human OSCAR ESCs cultured in self-renewal media (FGF2) and of ESC-derived EBs (Diff.) confirmed a marked decrease of RSL24D1 expression upon differentiation (Fig. 1f) and in adult human tissues (Supplemental Fig. S1f)[31]. In addition, RSL24D1 was expressed at similar levels in both human OSCAR and H9 ESCs maintained either in primed (TL and FGF2) or naive-like (TL2i) conditions[31] (Supplemental Fig. S1g). Taken together, these results indicate that RSL24D1 enhanced expression pattern is associated to pluripotency and evolutionarily conserved in PSCs.

### RSL24D1 is associated with nuclear pre-ribosomes in ESCs
The role of RSL24D1 in higher eukaryotes remains unknown. Therefore, to gain insight into its molecular functions in mouse ESCs, we first compared its sequence and structure with conserved homologs. Indeed, Rlp24, the yeast homolog of RSL24D1, is a RBF involved in the export of nuclear pre-60S ribosomal particles to the cytoplasm where they undergo the final steps of maturation, including Rlp24 substitution by the canonical RP eL24 (Rpl24)[16,32]. Multiple protein alignments of yeast Rlp24 to higher eukaryote homologs revealed that the first 130 amino acids of Rlp24 are well conserved

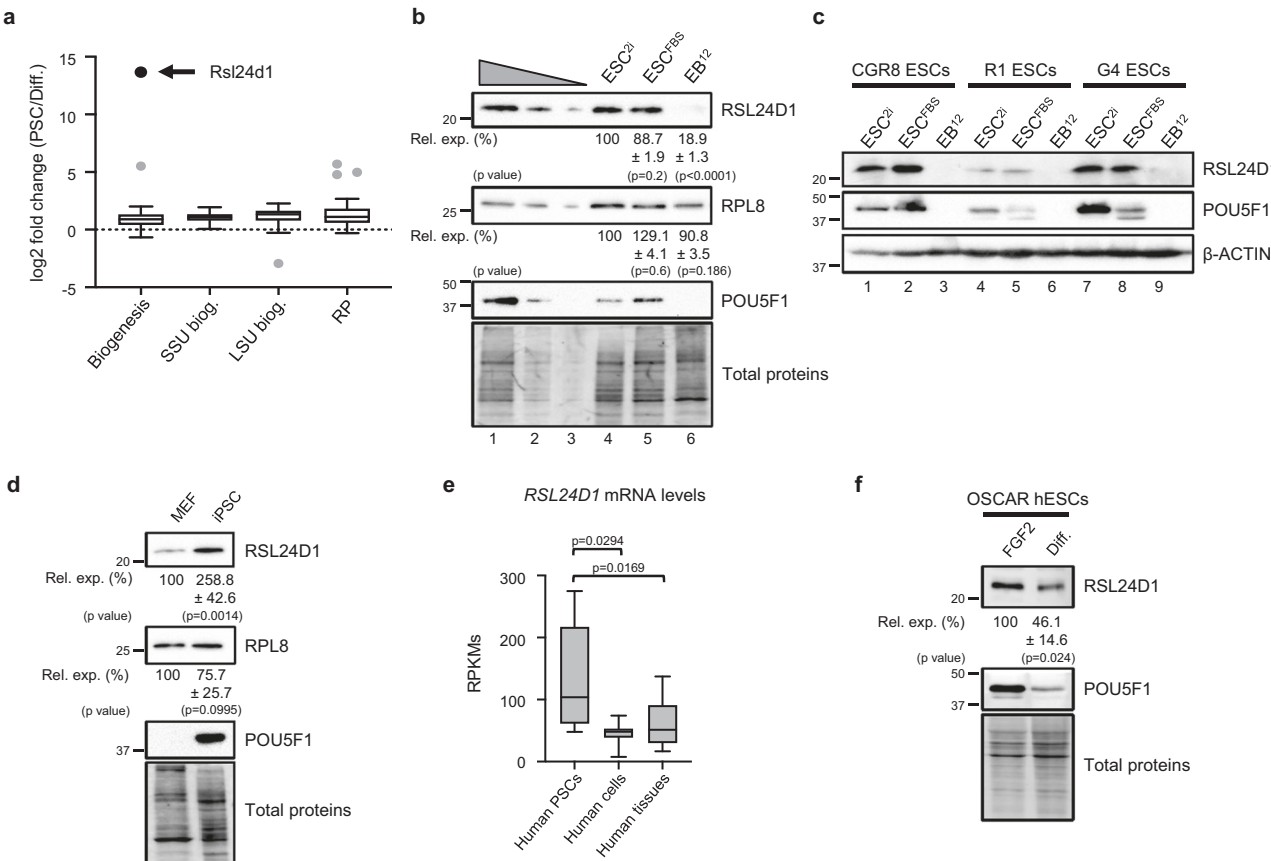

**Fig. 1 | Rsl24d1 expression is enriched in PSCs. a** Box plot representation of mRNA expression changes in mouse PSCs (5 ESC lines and 2 iPSC clones) and 6 differentiated cell lines, measured by RNA-seq, for factors involved in common ribosome biogenesis (Biogenesis), specific biogenesis factors of the 40S (SSU biog.) or 60S subunits (LSU biog.), and ribosomal proteins (RP). Outlier values and Rsl24d1 are represented by gray and black dots, respectively. Additional details are provided in Supplemental Fig. S1a and in Supplementary Data 1. The boxes represent the 25th to 75th inter-quartile (IQR) and the whiskers +/− 1.5 IQR. The median and the outliers are represented by a line and individual dots, respectively. **b** Representative immunoblots of mouse CGR8 ESCs cultivated in ESC$^{2i}$ or ESC$^{FBS}$ conditions, or differentiated as embryoid bodies for 12 days (EB$^{12}$) ($n = 4$). Lanes 1 to 3 correspond to serial dilutions of ESC$^{FBS}$ (1:1, 1:3, and 1:9, respectively). TCE labeling of proteins is used for normalization (Total Proteins). Quantifications of RSL24D1 and RPL8 signals normalized to total proteins and relative to the ESC$^{2i}$ condition (Rel. exp. (%)) are indicated. Two-tailed Student's $t$ test. **c** Representative immunoblots of CGR8, R1 and G4 mouse ESC lines cultured in ESC$^{2i}$, ESC$^{FBS}$ and differentiated (EB$^{12}$) conditions, as in panel (**b**). POU5F1 and β-ACTIN are shown as pluripotent

state and loading controls, respectively. **d** Representative immunoblots of MEFs and iPSCs reprogrammed from MEFs derived from R26$^{rtTA}$;Col1a1$^{4F2A}$ mice after ectopic expression of Pou5f1, Sox2, Klf4 and cMyc ($n = 4$). Quantifications of RSL24D1 and RPL8 immunoblot signals normalized to total proteins and relative to the MEF condition (Rel. exp. (%)) are indicated. Two-tailed Student's $t$ test. **e** Box plot representation of normalized *RSL24D1* mRNA levels in RPKMs (Reads per kilo base per million mapped reads) across independent samples of human PSCs ($n = 5$), cell lines ($n =$ )· and adult tissues ($n = 16$) based on published RNA-seq profiles[26]. Two-tailed Student's $t$ test. The median is indicated by a line, the boxes represent the 25th to 75th IQR, and the whiskers from the minimum to the maximum. **f** Representative immunoblots of human OSCAR ESCs maintained in pluripotent state in the presence of FGF2 or in vitro differentiated into EBs (Diff.). POU5F1 levels are indicative of the pluripotency state and TCE-labeled total proteins are used as a loading control ($n = 3$). Quantifications of RSL24D1 signals normalized to total proteins and relative to the FGF2 condition (Rel. exp. (%)) are indicated. Two tailed Student's $t$ test.

from yeast to human (Supplemental Fig. S2a, b). In particular, RSL24D1 homologs were strongly conserved in mammals, with a sequence identity over 96%, but lacked the yeast-specific C-terminal extension of Rlp24.

We next compared the predicted structure of mouse RSL24D1 with cryo-EM structures of yeast and human protein homologs, which have been recently obtained from nuclear pre-60S intermediates, while they are absent in cytoplasmic pre-60S particles[33,34]. Interestingly, the first 135 amino acids of mouse RSL24D1 almost perfectly matched the yeast Rlp24 and human RSL24D1 structures from pre-60S intermediates (Fig. 2a). Altogether, these results strongly support a conserved function of RSL24D1 in the nuclear maturation of the pre-60S particles in mouse ESCs.

To test this hypothesis, the localization of RSL24D1 was analyzed in CGR8 ESCs. Since no difference in RSL24D1 expression was observed between the ESC$^{2i}$ and ESC$^{FBS}$ conditions, all following experiments

were performed in CGR8 ESC$^{FBS}$. Cell fractionation assays confirmed that RSL24D1 was predominantly detected in the nuclear fraction (68%) and to a lower extent in the cytoplasmic fraction (32%), while RPL24 was almost exclusively present in the cytoplasm (95%) (Supplemental Fig. S2c). Furthermore, RSL24D1 was expressed in all colony-forming ESCs regardless of POU5F1 steady-state levels (Supplemental Fig. S2d) and was predominantly concentrated within nuclear foci containing the RBF Fibrillarin (FBL) (Fig. 2b), therefore suggesting that RSL24D1 is mostly located in ESC nucleoli[35].

We then asked whether RSL24D1 was associated with pre-ribosomal particles in CGR8 ESC$^{FBS}$. Following cell fractionation, pre-ribosomal and ribosomal particles were respectively isolated from nuclear and cytoplasmic fractions by ultracentrifugation on sucrose cushions. In contrast to RPL8 (LSU) and RPS6 (SSU) that were both present in nuclear and cytoplasmic ribosomal particles, RPL24 was exclusively present in cytoplasmic ribosomes (Fig. 2c). In addition, the

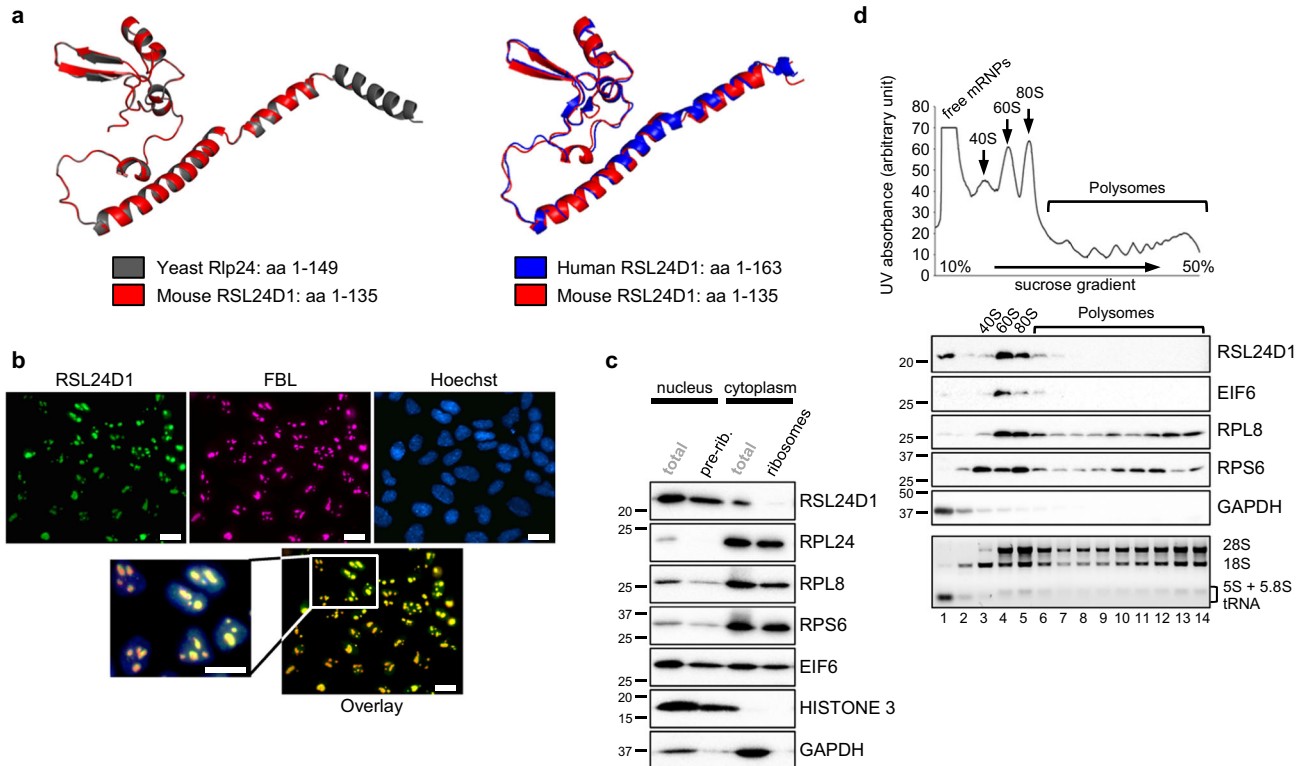

**Fig. 2 | RSL24D1 is associated with pre-60S subunits in mouse ESCs.**
**a** Comparison of the mouse RSL24D1 predicted structure (amino acids 1–135, red color) with the yeast Rlp24 structure (amino acids 1–149, gray color, left panel) and the human RSL24D1 structure (amino acids 1–163, blue color, right panel)[33,34]. The mouse RSL24D1 protein structure was predicted from the 6N8J and 6LSS PDB structures using the swiss-model structure assessment tool[69]. **b** Representative images of naive CGR8 cells stained with Hoechst and with anti-FBL and anti-RSL24D1 antibodies (20× objective) (n = 3). The scale bar represents 10 μm. **c** Representative immunoblots of ESC[FBS] nucleo-cytoplasmic fractions before (total) and after sucrose cushion purifications of nuclear pre-ribosomes (pre-rib.)

and cytoplasmic ribosomes (ribosomes) (n = 3). HISTONE 3 and GAPDH are shown as specific nuclear and cytoplasmic proteins, respectively. **d** Polysome profiling of CGR8 ESC[FBS] cytoplasmic extracts. Ribosome-free fractions (free mRNPs), 40S, 60S, 80S monosomes and polysomes are detected by UV-absorbance and indicated on the absorbance curve (upper panel). Representative immunoblots of gradient fractions (middle panel) (n = 3). GAPDH is used as a control of the free mRNPs. Total RNAs were extracted from collected fractions, analyzed on non-denaturing agarose gels, and revealed with ethidium bromide. The positions of the 28S, 18S, 5.8S, 5S rRNAs, and tRNAs are indicated (lower panel).

LSU biogenesis factor EIF6, homolog of yeast Tif6, co-purified with both nuclear and cytoplasmic ribosomal particles, whereas RSL24D1 was predominantly detected in nuclear pre-ribosomes and slightly associated with cytoplasmic ribosomes (Fig. 2c). These observations suggest that RSL24D1 is rapidly removed from cytoplasmic pre-ribosomes and most likely replaced by RPL24 after nuclear export, in agreement with observations in yeast and human[16,32,33].

To confirm that RSL24D1 is associated with pre-60S particles, cytoplasmic fractions were analyzed by polysome profiling assays to separate the 40S, 60S, 80S (monosomes) and polysomes (Fig. 2d). As expected, RPL8 and RPS6 were predominantly detected in 60S and 40S ribosomal fractions, respectively, and both proteins were also present in monosomes and polysomes (Fig. 2d). In contrast, EIF6 was mainly detected in 60S fractions (lane 4), suggesting that they also contain pre-60S particles, and to a lower extent in 80S fractions (lanes 5 and 6), most likely reflecting an incomplete separation of 80S and 60S. Strikingly, RSL24D1 was only detected in EIF6-containing fractions. Altogether, these results indicate for that RSL24D1 is a LSU RBF in higher eukaryotes, which is incorporated into nucleolar pre-60S particles, transits to the cytoplasm and is subsequently removed from cytoplasmic pre-60S.

## RSL24D1 loss impairs ribosome biogenesis and translation

To confirm that RSL24D1 is involved in ribosome biogenesis, we then determined whether its depletion impacts the accumulation and activity of mature cytoplasmic ribosomes. Rsl24d1 siRNA treatment for

72 h resulted in an efficient depletion of RSL24D1 (>67%) in CGR8 ESCs compared to control non-targeting siRNAs (Fig. 3a). Interestingly, RSL24D1 depletion did not detectably affect the structure and the number of FBL-containing nucleoli (Supplemental Fig. 3a), thus suggesting that decreasing RSL24D1 levels does not significantly impair early ribosome biogenesis.

Next, the effect of RSL24D1 depletion on ribosome production was assessed by polysome profiling assays conducted on control- or Rsl24d1-siRNA treated ESCs. The transient depletion of RSL24D1 caused a significant loss of 60S and 80S relative to 40S subunits (Fig. 3b). To achieve a more stable depletion of RSL24D1, we designed two independent shRNAs targeting *Rsl24d1* mRNAs (sh-Rsl24d1-1 and sh-Rsl24d1-2), which resulted in a 51 and 93% depletion of RSL24D1, respectively, compared to control shRNAs (Supplemental Fig. S3b). Interestingly, expressing sh-Rsl24d1-2, which provided the most robust silencing of RSL24D1, also impaired the accumulation of the 80S and 60S subunits, to a similar extent than siRNA-mediated depletion (Supplemental Fig. S3c, d). To confirm that the alteration of ribosomal subunit accumulation was directly caused by RSL24D1 downregulation, we established a CGR8 rescue cell line expressing a siRNA-resistant Rsl24d1-encoding mRNA in a doxycycline inducible manner (Supplemental Fig. S3e). Strikingly, RSL24D1 partial rescue in ESCs was sufficient to prevent a loss of 80S, and to a lower extent, of 60S (Fig. 3c).

To further characterize the molecular alterations resulting from RSL24D1 depletion, nuclear and cytoplasmic ribosomal fractions were isolated from si-CTL and si-Rsl24d1-treated ESCs (Fig. 3d). The

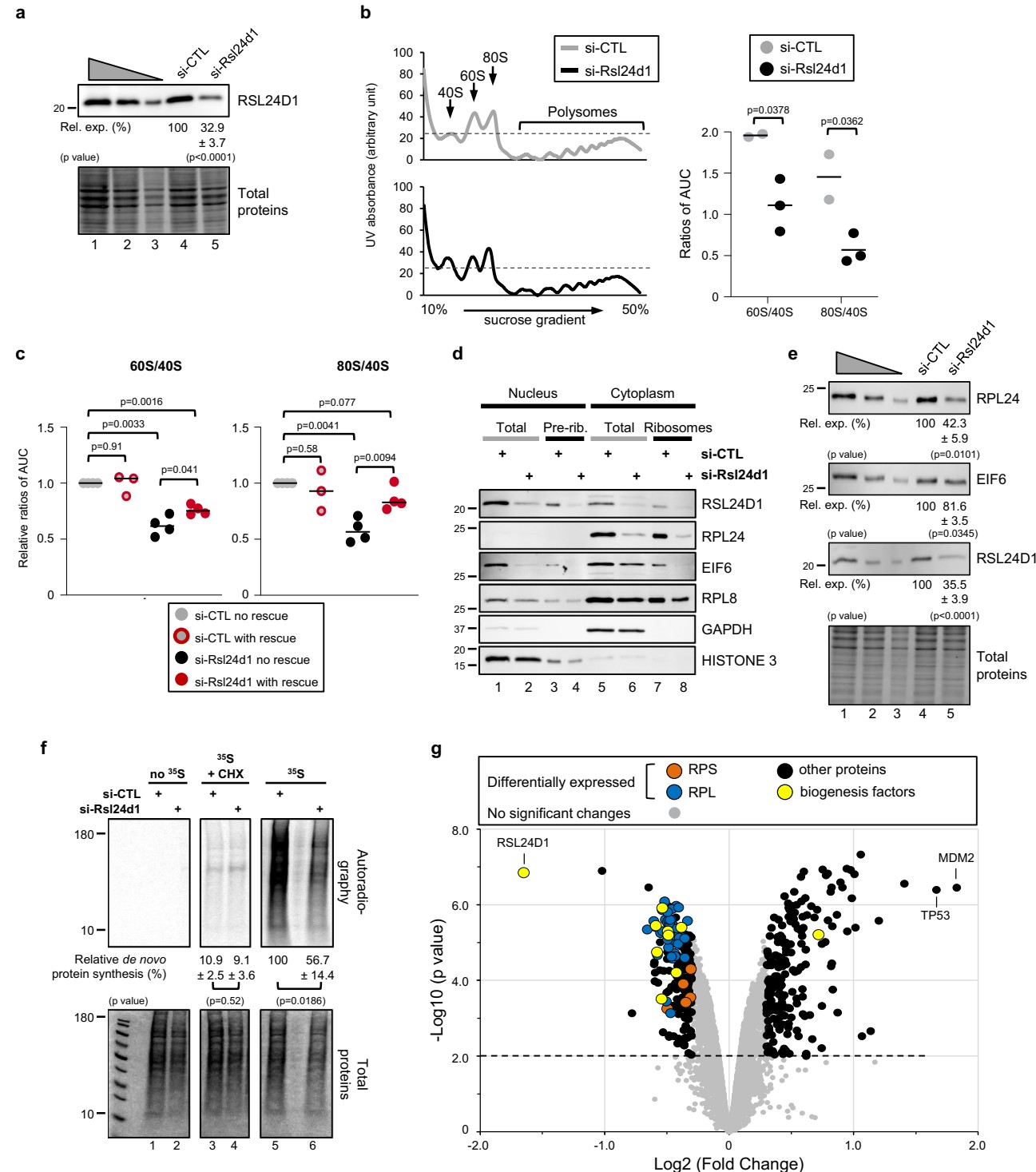

depletion of RSL24D1 resulted in a significant loss of EIF6 association with nuclear pre-ribosomes (lanes 3–4) and cytoplasmic particles (lanes 7–8). RPL24 inclusion into cytoplasmic ribosomal particles was also strongly impaired upon RSL24D1 knockdown (lanes 7–8), while the presence of the canonical RPL8 in both nuclear pre-ribosomes and cytoplasmic ribosomes was only marginally affected (lanes 1–8). Moreover, the consistent decrease of RPL24 and EIF6 expression in cytoplasmic (lanes 5–6) and nuclear (lanes 1–2) fractions upon RSL24D1 depletion was further confirmed in total extracts (Fig. 3e). These observations suggest that defects in EIF6 and RPL24 association with pre-60S particles may either result from a recruitment defect or from a decrease in the steady-state levels of these proteins caused by

stability or translation alterations. Altogether, these results indicate that RSL24D1 is required for the bona fide maturation of pre-60S particles, at least by allowing EIF6 and RPL24 association with nuclear pre-ribosomes and mature cytoplasmic ribosomes, respectively. Hence, RSL24D1 plays a critical role in the production of mature LSU in mouse ESCs.

Finally, we hypothesized that RSL24D1 depletion may also affect global protein synthesis in ESCs. $^{35}$S pulse-chase labeling assays were performed on CGR8 ESCs transfected with si-CTL or si-Rsl24d1, in the presence or absence of cycloheximide (CHX), an inhibitor of translation elongation (Fig. 3f). As expected, CHX almost completely abrogated $^{35}$S incorporation in newly synthetized proteins (lanes 3–4),

**Fig. 3 | RSL24D1 depletion alters ribosome biogenesis and protein translation. a** Representative immunoblot of total extracts from si-CTL and si-Rsl24d1 treated CGR8s ($n = 3$). Lanes 1 to 3 correspond to serial dilutions of ESC^FBS (1:1, 1:3, and 1:9, respectively). Quantifications of RSL24D1 signals normalized to total proteins and relative to the si-CTL condition (Rel. exp. (%)) are indicated. Two tailed Student's $t$ test. **b** Left panel. Polysome profiling of cytoplasmic extracts from ESC^FBS treated with si-CTL or si-Rsl24d1. 40S, 60S, 80S, and polysomes are indicated. Right panel. Dot plot indicating ratios of 60S/40S and 80S/40S absorbance peaks calculated by determining the area under the curve (AUC) for the 40S, 60S, and 80S absorbance signals ($n = 3$). Two tailed Student's $t$ test. **c** Dot plots indicating ratios of 60S/40S and 80S/40S absorbance peaks, as previously described in panel (**b**), in CGR8-RESCUE cells treated with si-CTL and cultured in the absence (si-CTL no RESCUE) or presence of doxycycline (si-CTL with RESCUE), or treated with si-RSL24D1 while expressing (si-Rsl24d1 with RESCUE) or not (si-Rsl24d1 no RESCUE) the ectopic RSL24D1 protein ($n = 3$). Two tailed Student's $t$ test. **d** Representative immunoblots of nucleo-cytoplasmic fractions of CGR8 ESC^FBS transfected with si-CTL or si-Rsl24d1, before (Total) and after ribosome purification on sucrose cushions (pre-rib. or ribosomes) ($n = 3$). HISTONE 3 and GAPDH are shown as specific nuclear and cytoplasmic proteins, respectively. **e** Representative immunoblots of total extracts from si-CTL and si-Rsl24d1 treated CGR8 ESC^FBS ($n = 3$). Lanes 1 to 3 correspond to serial dilutions of si-CTL-treated ESC^FBS (1:1, 1:3, and 1:9, respectively). Quantifications of immunoblot signals normalized to total proteins and relative to si-CTL-treated cells (Rel. exp. (%)) are indicated. Two tailed Student's $t$ test. **f** Representative autoradiography (upper panel) and coomassie staining (lower panel) of SDS-PAGE from si-CTL and si-Rsl24d1 treated ESC^FBS, in the presence or absence of cycloheximide (CHX) and ³⁵S-labeled methionine and cysteine. Quantifications of ³⁵S signals (autoradiography) are normalized to total proteins (coomassie staining) and expressed relative to si-CTL- and ³⁵S-treated ESC^FBS ($n = 3$). Two tailed Student's $t$ test. **g** Volcano plot representation of the distribution of protein expression changes detected by mass spectrometry in si-CTL and si-Rsl24d1 treated CGR8s and detailed in Supplementary Data 2. Different protein categories are highlighted with specific colors.

while RSL24D1 depletion caused a significant reduction (43%) of de novo protein synthesis activity in ESCs (lanes 5–6). Moreover, RSL24D1 expression rescue was sufficient to significantly restore de novo protein synthesis in ESC^FBS (Supplemental Fig. S3f).

To define the role of RSL24D1 on ESC proteome regulation, a mass-spectrometry-based quantitative proteomic analysis was performed on si-CTL or si-Rsl24d1 treated cells (Supplementary Data 2). These results revealed that steady-state expression levels of 391 out of 6711 detected proteins were altered when RSL24D1 was depleted (Fig. 3g and Supplemental Fig. S3g). 192 proteins were downregulated (49,1%), including several RBFs of the pre-60S particles and most of the LSU RPs (46/48), while only 6 out of 33 RPs of the 40S were affected (Supplemental Fig. S3h). Downregulated proteins were strongly enriched in annotations associated with "ribosome" ($p < 9E–18$) and "translation" ($p < 2E–08$) (Supplemental Fig. S3i and Supplementary Data 3A). Conversely, 199 proteins (50.1%) showed increased expression and were enriched in functions associated with "negative regulation of cell growth" ($p < 1E–03$) and "apoptosis" ($p < 1E–03$) (Supplemental Fig. S3i and Supplementary Data 3B). Interestingly, the 2 most upregulated proteins were MDM2 and TP53, therefore strongly supporting that RSL24D1 depletion and subsequent LSU alterations activate the ribosome biogenesis quality control pathway, which transduces ribosomal stress by stabilizing p53[36]. Altogether, these results convincingly demonstrate that RSL24D1 is an essential LSU RBF required for the steady-state protein synthesis in mouse ESCs.

## Rsl24d1 maintains pluripotent transcriptional programs

The effects of *Rsl24d1* knockdown on ribosome biogenesis and translation are likely to impair the regulation of specific gene programs in mouse ESCs. To address this question, RNA-Seq profiling was performed on CGR8 ESCs transfected with control- or Rsl24d1-targeting siRNAs, without or with a rescue expression of ectopic RSL24D1. Genes with the most significant mRNA expression changes (fold change >1.8; $p < 0.01$) were further analyzed. RSL24D1 loss resulted in the altered expression of 1404 genes, including 1099 upregulated (78%) and 305 downregulated (22%) genes (Supplementary Data 4A).

A Gene Ontology analysis of genes decreased upon *Rsl24d1* depletion revealed a significant enrichment in terms associated with metabolic and transport processes ($p < 0.01$) (Fig. 4a; Supplementary Data 5A). Conversely, upregulated genes upon si-Rsl24d1 treatment were strongly associated with "developmental processes", "cell differentiation", "proliferation" and "signal transduction" annotations ($p < 1E–06$) (Fig. 4a; Supplementary Data 5B). Strikingly, RSL24D1 expression rescue considerably reduced both the number of differentially expressed genes, with 444 upregulated and 86 downregulated genes respectively (Supplemental Fig. S4a) (Supplementary Data 4B) and the amplitude of expression changes observed in this condition (Fig. 4b), specially for germ layer marker genes (Supplemental

Fig. S4b). Along this line, differentially expressed genes in rescue ESCs (Supplemental Fig. S4c) have a less significant association to GO annotations previously shown to be enriched as a result of RSL24D1 depletion (Fig. 4a) (Supplementary Data 5C, D). These results suggest that *Rsl24d1* expression in mouse ESCs is required to maintain a coordinated regulation of specific transcription programs that control important ESC biological processes.

We next hypothesized that the alterations of ESC transcription programs caused by *Rsl24d1* depletion could result from an impaired expression of key transcriptional or epigenetic stemness regulators[37]. Indeed, an analysis using the StemChecker algorithm[38] revealed that genes downregulated ($n = 305$) upon *Rsl24d1* mRNA knockdown were enriched in targets of key PTFs, including *Nanog*, *Pou5f1*, *Klf4*, and *Sox2* (Fig. 4c; Supplementary Data 6A). Conversely, upregulated genes ($n = 1099$) were preferentially enriched in targets of essential epigenetic regulators of the polycomb family, including *Suz12*, *Eed* and *Ezh2* from PRC2, and *Rnf2* from PRC1 (Fig. 4c; Supplementary Data 6B). As expected, RSL24D1 expression rescue strongly reduced the scores of these target gene predictions (Supplemental Fig. S4d), further reflecting a less profound impairment of ESC transcriptome in the rescue ESCs compared to RSL24D1-depleted cells.

To establish whether genes differentially expressed upon *Rsl24d1* knockdown are direct targets of these PTFs and PRC factors, we next examined the promoter regions of these 1404 genes for enrichment in binding sites for these factors, using ChIP-seq assays performed in mouse ESCs (ChIP-Atlas database[39]) (Fig. 4d and Supplementary Data 7). ChIP-seq data revealed a slight enrichment of SOX2 binding sites near the promoters of downregulated genes, yet POU5F1 and NANOG were not preferentially associated to these promoters. However, the binding sites for these 3 PTFs were less frequently detected in the promoter regions of genes upregulated in absence of Rsl24d1. Conversely, the promoter regions of upregulated genes were significantly enriched in binding sites for PRC1 and PRC2 factors compared to those of downregulated genes (Fig. 4d). Strikingly, about half of promoter regions of upregulated genes were bound by EZH2 (36.8%), SUZ12 (42.4%), EED (23.8%) or RNF2 (46.5%) in mouse ESCs (Fig. 4d and Supplementary Data 7), and more than a third of upregulated genes were recently defined as PRC2 target genes (Fig. 4e)[8]. These results further confirmed that genes down- or upregulated upon RSL24D1 depletion are enriched in direct targets of POU5F1/NANOG or EZH2/SUZ12/EED/RNF2, respectively.

PRC2 is a key epigenetic repressor responsible for H3K27me2 and H3K27me3 modifications, which controls the expression of early commitment genes in ESCs[2,5–7]. Since a large proportion of differentially expressed genes upon RSL24D1 depletion were associated with PRC2 binding sites, we next analyzed the H3K27 methylation status of their promoter regions using published dataset obtained from mouse ESCs[7] (Supplemental Fig. S4e). While the promoters of downregulated

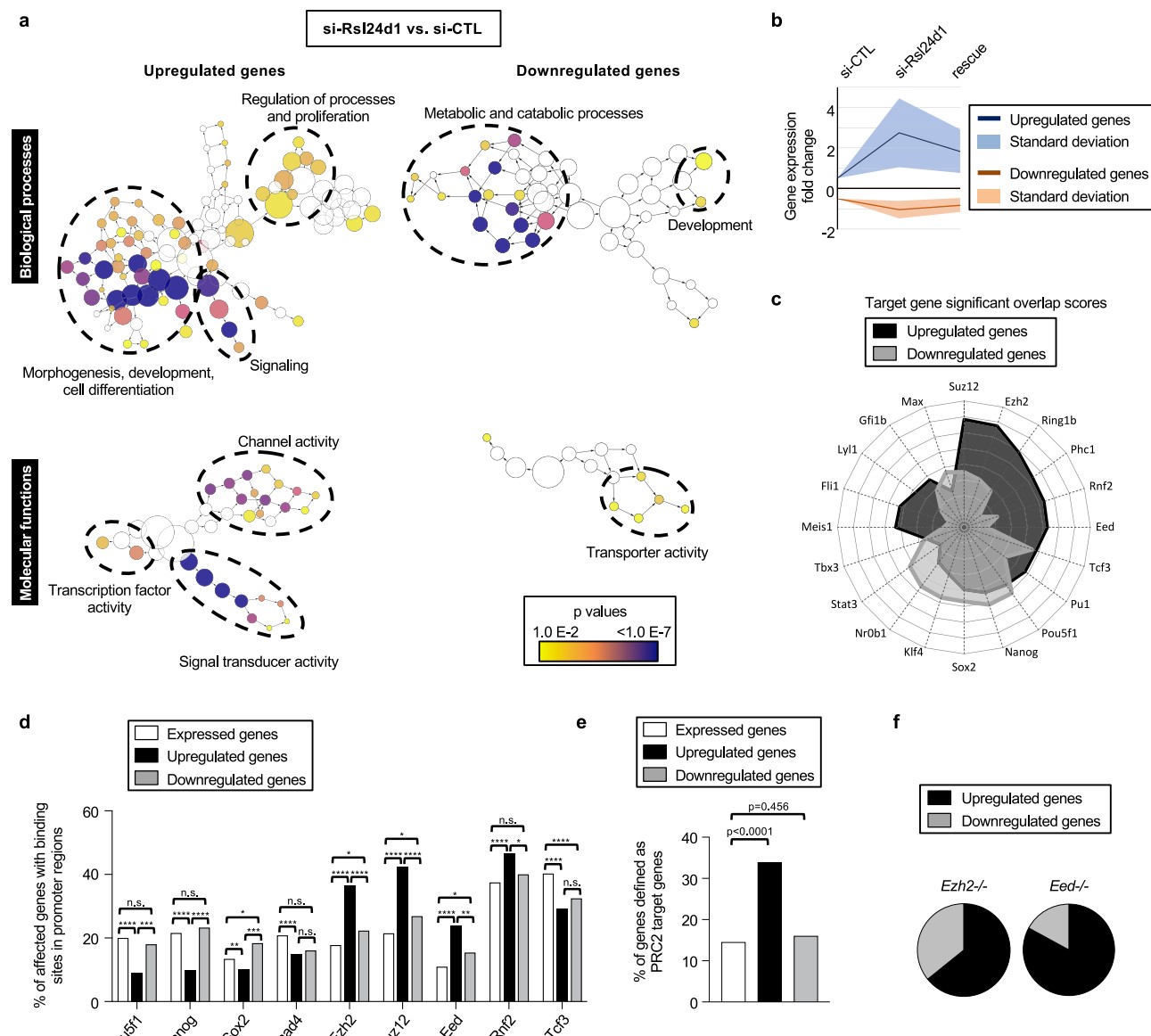

**Fig. 4 | RSL24D1 is required to maintain the regulation of pluripotency and differentiation transcriptional programs. a** Gene Ontology (GO) enrichment analyses of biological process and molecular function terms associated with genes differentially expressed in CGR8 cells treated with Rsl24d1 siRNAs. The left and right panels represent hierarchical trees of the most enriched terms for genes up- or downregulated in Rsl24d1-depleted cells, respectively. The size of the nodes represents the numbers of genes associated to each GO term, and the corresponding *p* values are indicated by color codes. The most represented GO term categories are indicated. The corresponding data are available in Supplementary Data 5A, B. **b** Graph showing the average fold change for the 200 most differentially expressed genes between si-CTL, si-Rsl24d1 (si-Rsl24d1 no rescue) and rescue (si-Rsl24d1 with rescue) conditions. **c** Radar plot summarizing the StemChecker

analysis for PTFs and chromatin-associated factors with a significant association score to the differentially regulated genes in si-Rsl24d1 cells. The corresponding data are available in Supplementary Data 6A, B. **d** Histogram showing the relative proportion of expressed genes (DEseq2 corrected *p* values <0.01, *n* = 6690), up- (*n* = 1099) and downregulated (*n* = 305) genes in Rsl24d1-depleted CGR8 cells containing binding sites for indicated transcription and chromatin-associated factors in their promoter region, established by ChIP-seq analyses[77]. Two sided Fisher's exact test: n.s. *p* > 0.05, **p* < 0.05, ***p* < 0.01, ****p* < 0.001, *****p* < 0.0001. **e** Histogram representing the relative proportions of up- and downregulated genes that are known PRC2 target genes[8]. Two sided Fisher's exact test. **f** Analysis of the proportion of genes affected in Rsl24d1-depleted CGR8 cells displaying similar expression changes in *Eed*[-/-] or *Ezh2*[-/-] ESCs[7,9].

genes were increasingly associated to all H3K27me marks compared to a set of 6690 detected genes, those of upregulated genes showed the strongest enrichment in H3K27me3 repressive marks. These results suggest that the levels of H3K27me3, and of H3K27me2 to a lower extent, may be altered when RSL24D1 is depleted. Accordingly, si-Rsl24d1 treated CGR8 cells displayed lower levels of nuclear H3K27me3 compared to si-CTL treated cells (Supplemental Fig. S4f), suggesting that some of the alterations in gene expression caused by RSL24D1 loss may directly result from this global reduction of H3K27me3.

Finally, we compared RNA-seq predictions from si-Rsl24d1 ESCs with expression profiling from EED and EZH2 knockout mouse ESCs[5,6]. Strikingly, genes similarly impaired in PRC2 mutant ESCs and RSL24D1-depleted ESCs were predominantly upregulated genes (Fig. 4f). Altogether, these results suggest that genes up- or downregulated upon RSL24D1 knockdown are likely controlled by distinct molecular mechanisms. On the one hand, downregulated genes are enriched in key PTF targets, suggesting that the transcriptional regulatory activities of POU5F1 and NANOG are decreased in si-Rsl24d1 treated ESCs. On the other hand, the enrichment of H3K27me3 sites and binding

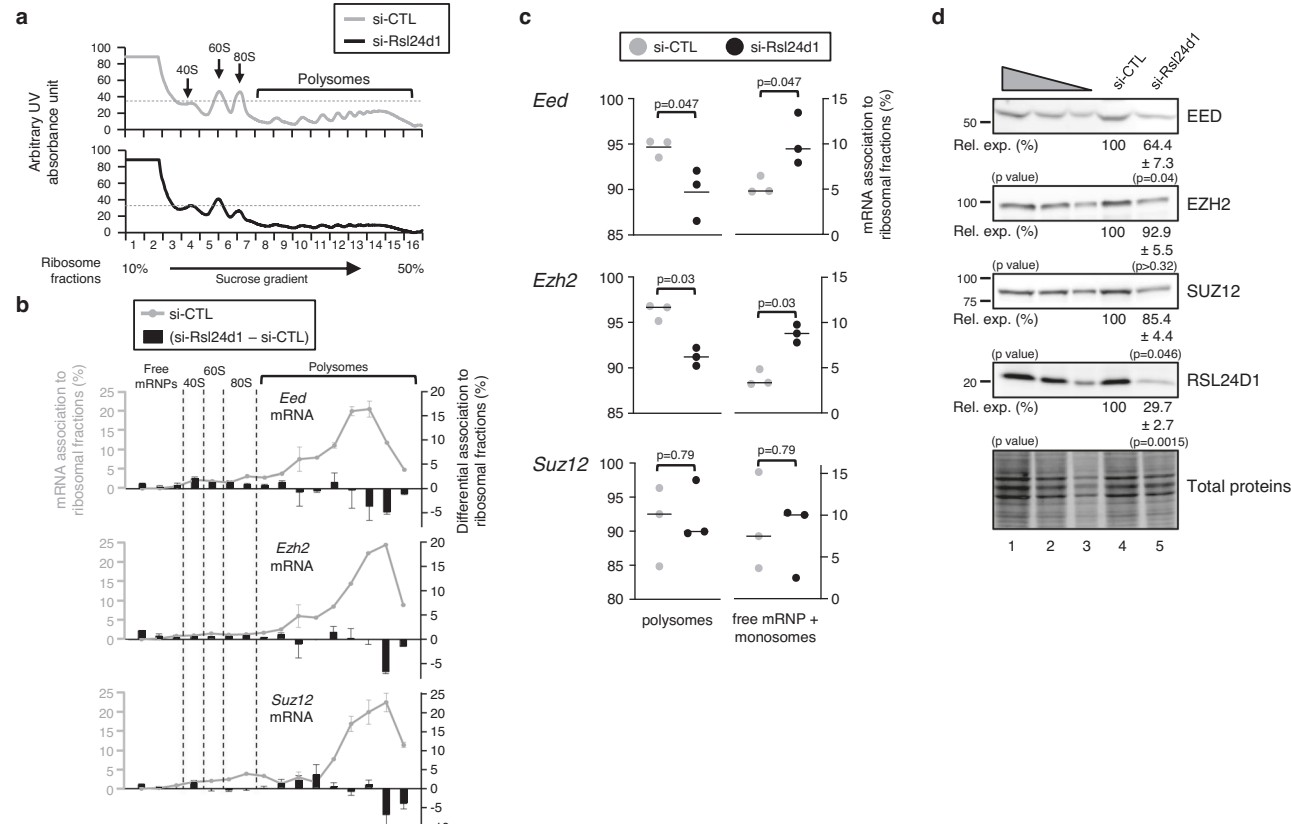

**Fig. 5 | RSL24D1 depletion impairs the translation of core PRC2 factors.**
**a** Polysome profiling obtained after centrifugation on sucrose gradients of cyto-plasmic extracts from naïve pluripotent CGR8 cells treated with non-targeting (si-CTL, gray) or Rsl24d1-targeting siRNAs (si-Rsl24d1, black). 40S, 60S, 80S monosomes and polysomes are detected by UV-absorbance and indicated on the absorbance curve. **b** Graphs representing *Eed*, *Ezh2,* and *Suz12* mRNA levels mea-sured by RT-qPCR in each fraction collected from three independent polysome profiling assays (gray curves, left scales) after normalization with a Luciferase spike-in mRNA and relative to the total amount of mRNAs detected in the polysome profiling (sum of all fractions). The bar graph represents the differential detection of each mRNA in each fraction in CGR8 cells treated with si-Rsl24d1 and control

siRNAs (black bars, right scales). **c** Dot plots representing the relative proportions of *Eed*, *Ezh2*, and *Suz12* mRNAs detected by qRT-PCR in free mRNPs/monosomal or polysomal fractions, based on quantifications detailed in the Fig. 5b, in si-CTL (gray) or si-Rsl24d1 (black) treated CGR8 ESC^FBS (*n* = 3). Two tailed Student's *t* test. **d** Representative immunoblots of RSL24D1 (*n* = 3), EED (*n* = 3), EZH2 (*n* = 3) and SUZ12 (*n* = 4) in si-CTL or si-Rsl24d1 treated CGR8 ESC^FBS. Lanes 1 to 3 correspond to serial dilutions of si-CTL-treated ESC^FBS (1:1, 1:3, and 1:9, respectively). TCE-labeled total proteins are used for normalization. Quantifications of immunoblot signals normalized to total proteins and relative to the si-CTL-treated conditions (Rel. exp. (%)) are indicated. Two tailed Student's *t* test.

sites for PRC2 proteins in promoter regions of upregulated genes suggests that RSL24D1 depletions most likely hinder the repressive activity of PRC2, therefore, leading to both reduced H3K27me3 deposition and aberrant premature activation of early developmental genes (Supplemental Fig. S4b).

**RSL24D1 loss impairs of the translation of key stemness factors**
To investigate how RSL24D1 modulates the activity of PTFs and PRC2 factors, we first measured whether the loss of RSL24D1 affected their transcription. The mRNA levels of these PTFs and PRC factors were slightly impaired in si-Rsl24d1 ESCs, except for *NrOb1* that was strongly affected (Supplemental Fig. S5a), suggesting that changes of mRNA expression alone may not be sufficient to account for the alterations of gene expression program observed in ESCs. Since RSL24D1 loss decreased global protein synthesis, the alteration of PTFs and PRC2 factors could also result from a perturbation of the translation of their corresponding mRNAs.

Thus, we next analyzed the association of *Eed*, *Ezh2*, and *Suz12* mRNAs with the different ribosomal fractions in CGR8 cells treated with either si-CTL or si-Rsl24d1 (Fig. 5a). As expected, the majority (>92%) of *Eed*, *Ezh2* and *Suz12* mRNAs were associated with polysomes and therefore actively translated in ESC^FBS (Fig. 5b, gray curves). However, RSL24D1 depletion significantly reduced the association of

*Eed* and *Ezh2* mRNAs with polysomal fractions while increasing their detection in free mRNPs and monosomes (Fig. 5b, black bars, and Fig. 5c). Although the switch of *Suz12* mRNAs from polysomal to monosomal fractions upon RSL24D1 depletion was not statistically significant, analyses of the polysome fractions alone seemed to indi-cate that *Suz12* mRNAs tend to transit from heavy to light polysomes in si-Rsl24d1 treated ESCs (Fig. 5b).

To further confirm that RSL24D1 downregulation impaired the translation of *Suz12*, *Eed* and *Ezh2* mRNAs, steady-state expression levels of corresponding proteins were assessed by semi-quantitative western blots on CGR8 total extracts. Interestingly, RSL24D1 depletion induced a significant reduction of EED and SUZ12 levels, albeit no significant downregulation was observed for EZH2 (Fig. 5d). Alto-gether, these observations suggest that the loss of PRC2 activity upon RSL24D1 depletion is likely caused by a defect in the translation of *Eed*, *Ezh2* and *Suz12* mRNAs in ESCs.

Reminiscent of what was observed for PRC2 factors, we hypo-thesized that a decrease in *Pou5f1* and *Nanog* translation may be responsible for the observed downregulation of POU5F1 and NANOG target genes in RSL24D1-depleted CGR8 cells. The association of *Nanog* and *Pou5f1* mRNAs to both monosomes and polysomes was also analyzed, as described above (Supplemental Fig. S5b). Although RSL24D1 depletion caused a significant shift of *Nanog* mRNAs from

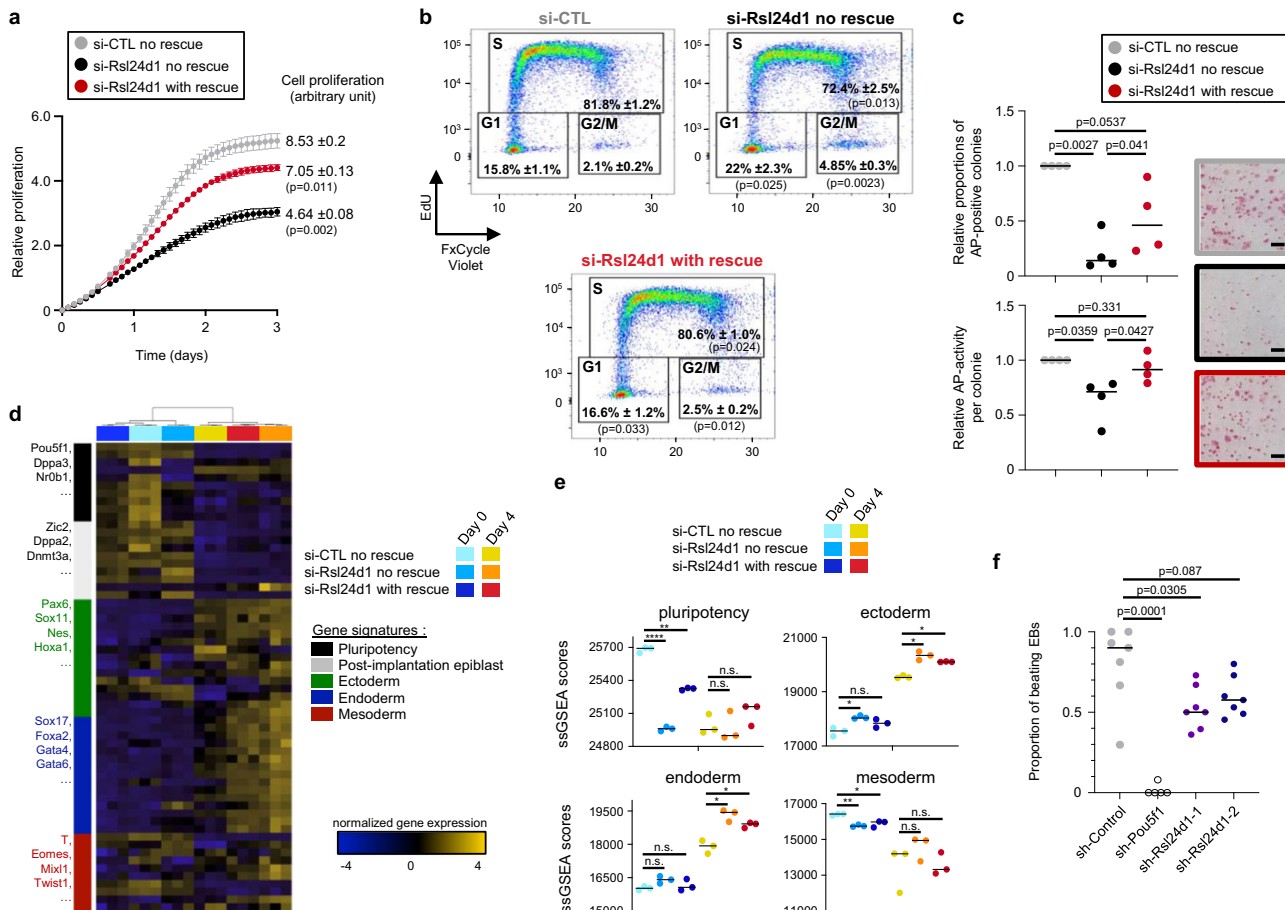

**Fig. 6 | RSL24D1 is required for mouse ESC proliferation and self-renewal.**
**a** Analysis of proliferative capacities defined by the cell confluence (Incucyte® technology) of CGR8 ESC^FBS treated with si-CTL or si-Rsl24d1, without or with doxycycline-induced RSL24D1 rescue (*n* = 3). Data are presented as mean values +/− SEM. Multiple paired *t* test. **b** FACS analysis of EdU incorporation and DNA content (FXCycle Violet) in ESC^FBS treated with si-CTL or si-Rsl24d1, with or without RSL24D1 protein rescue. The percentage of cells in the different phases of cell cycle are indicated. Two tailed Student's *t* test. **c** Analysis of the self-renewing capacities of CGR8 ESC^FBS estimated by the quantification of the number of individual colonies with alkaline phosphatase activity detected by colorimetric labeling (upper graph) and the relative alkaline phosphatase activity in each colony (lower graph) (*n* = 4). Representative images of the ESC colonies are shown for the si-CTL (gray frame), si-

Rsl24d1 without (black frame) or with (red frame) RSL24D1 protein rescue conditions. Two tailed Student's *t* test. Scale bars represent 400 μm. **d** Heatmap representation of the expression profiles of 60 marker genes in si-CTL, si-Rsl24d1, and si-Rsl24d1-rescue CGR8 cells cultured with LIF (Day 0) or in absence of LIF with retinoic acid for 4 days to induce differentiation (Day 4). **e** Dot plots representing pluripotency, ectoderm, endoderm, and mesoderm gene signatures, as previously defined[43]. Each biological replicate (*n* = 3) is indicated by an individual dot and the average score for each condition by a bar. Multiple paired *t* test: n.s. *p* > 0.05, \**p* < 0.05, \*\**p* < 0.01, \*\*\*\**p* < 0.0001. **f** Analysis of the proportion of 12-day old EBs, generated from CGR8 cells expressing control- (*n* = 7), Pou5f1- (*n* = 5) or Rsl24d1-shRNAs (*n* = 7), containing functional self-beating cardiomyocytes. Two tailed Student's *t* test.

polysomes to monosomes fractions (Supplemental Fig. S5b, c), *Pou5f1* mRNAs association to these ribosome fractions was not significantly impaired. Furthermore, semi-quantitative western blots demonstrated a significant reduction in both POU5F1 and NANOG protein levels in si-Rsl24d1 treated cells compared to si-CTL cells (Fig. S5d). Finally, immunostaining experiments confirmed that POU5F1 downregulation in si-Rsl24d1 treated ESCs did not result from the emergence of POU5F1-negative differentiated cells but rather from a global decrease in POU5F1 expression likely caused both by reduced expression and translation of *Pou5f1* mRNAs (Supplemental Fig. S5e). Altogether, these results confirm that RSL24D1 depletion directly impairs the translation and the accumulation of both key PTFs and core PRC2 factors, which respectively play pivotal roles in controlling gene expression programs and the chromatin landscape underlying cell fate decisions in ESCs.

### High RSL24D1 levels maintain ESC self-renewal and lineage commitment

Since transient RSL24D1 depletion altered the expression of several PTFs and promoted the activation of genes involved in developmental

programs, we examined the functions of RSL24D1 for ESC fundamental self-renewal and differentiation capacities. First, siRNA-mediated RSL24D1 downregulation significantly impaired the proliferation of CGR8 cells (Fig. 6a), while rescuing RSL24D1 was sufficient to limit the proliferation alteration. Similarly, the constitutive expression of Rsl24d1-targeting shRNAs altered the kinetics of proliferation of CGR8 cells in a dose-dependent manner (Supplemental Fig. 6a), thus suggesting that RSL24D1 expression levels are directly correlated with cell proliferation. To this end, we also analyzed the impact of RSL24D1 loss on the proliferative capacities of mouse cell lines expressing different levels of RSL24D1. Compared to ESC^FBS, MEF and NIH3T3 cells expressed reduced levels of endogenous RSL24D1, respectively 16 and 45% (Supplemental Fig. 6b). While the loss of RSL24D1 in CGR8 cells led to a 46% decrease of proliferation, a similar si-RNA mediated depletion of RSL24D1 in MEFs and NIH3T3 reduced cell proliferation by 38 and 25%, respectively (Supplemental Fig. 6c, d). Altogether, these observations suggest that not only high RSL24D1 expression is required in ESCs to sustain enhanced proliferative capacities, but it may play a broader role in sustaining proliferation in non-pluripotent cells.

Interestingly, pluripotent mouse ESCs have a short G1 phase, as it is more prone to promote differentiation, and are predominantly observed in S phase[40,41]. Thus, we hypothesized that alterations of gene expression programs upon RSL24D1 depletion could correlate with alterations of the G1 phase in ESCs. To address this question, we next analyzed the distribution of CGR8 cells in the different cell cycle phases as RSL24D1 expression is altered (Supplemental Fig. 7). The majority (82%) of si-CTL CGR8 cells are in the S phase while about 16 and 2% are in the G1 and G2/M phases, respectively (Fig. 6b). RSL24D1 depletion caused a significant increase in the proportion of cells in G1 (22%) and G2/M (5%) phases with a concomitant reduction of cells in the S phase (72%) (Fig. 6b). This result perfectly mirrors changes in cell cycle distribution observed upon mouse ESC differentiation, which result from a lengthening of the G1 phase[40,41]. As expected, RSL24D1 expression rescue restored CGR8 cell cycle distribution, with about 81% of ESCs in S phase and only 17 and 2.5% in G1 and M/G2 phase, respectively.

The evidence of reduced POU5F1 and NANOG levels (Supplemental Fig. S5d) together with altered G1 phase led us to examine whether RSL24D1 depletion impacts ESC self-renewal. CGR8 ESCs were seeded at clonal density to recapitulate the formation of individual undifferentiated colonies displaying high levels of alkaline phosphate (AP) activity[42]. RSL24D1 siRNA-mediated depletion caused a significant decrease of both the number of AP-positive colonies (Fig. 6c upper panel) and the mean signal intensity directly correlated to the relative AP activity in each colony (Fig. 6c lower panel). Conversely, RSL24D1 expression rescue was sufficient to partially recover ESC self-renewal capacities (Fig. 6c). These results were confirmed by stable expression of non-targeting shRNAs (sh-Control), or shRNAs targeting *Pou5f1* or *Rsl24d1* mRNAs, as previously described. As expected, POU5F1 down-regulation led to a drastic reduction (>90%) in the number of AP-positive colonies compared to control shRNA-treated cells (Supplemental Fig. S6e). shRNA-mediated RSL24D1 loss caused a dose-dependent and significant reduction (>50%) of AP-positive ESC colonies (Supplemental Fig. S6e), similar to the siRNA-mediated depletion. Altogether, these results, therefore, suggest that a high expression of RSL24D1 is required to support self-renewal in mouse ESCs.

Finally, we assessed whether RSL24D1 depletion also altered lineage choice and differentiation capacities of ESC^FBS. si-CTL and si-RSL24D1 treated ESCs, without or with RSL24D1 rescue, were cultured in LIF-containing self-renewal media (day 0) or in absence of LIF and with retinoic acid (RA) for 4 consecutive days to induce ESC differentiation in vitro[43]. To characterize the molecular changes triggered by continuous RA exposure and RSL24D1 depletion, we analyzed global gene expression changes by RNA-seq. At day 0, this analysis confirmed that the rescued expression of RSL24D1 significantly alleviated the gene expression alteration induced by si-Rsl24d1 treatment (Supplemental Fig. S8a). Indeed, both individual pluripotency markers such as *Zfp42*, *Klf4*, and *NrOb1*, as well as a pluripotency signature previously described[43] showed higher expression in RSL24D1-rescued cells compared to si-Rsl24d1 treated ESCs (Fig. 6d, e, day 0).

In addition, a principal component analysis revealed that all samples treated with RA for 4 days cluster together, regardless of the expression status of RSL24D1 (Supplemental Fig. S8a). This result was further confirmed when analyzing the differentially expressed genes between RA- (day 4) and LIF-treated cells (day 0), which revealed similar transcriptome modifications (Supplemental Fig. S8b and Supplementary Data 4C–F) and GO annotation enrichments (Supplemental Fig. S8c and Supplementary Data 8A–C) in si-CTL, si-Rsl24d1 and RSL24D1 rescue samples. As previously described[43], RA treatment of si-CTL CGR8 cells triggered a strong downregulation of numerous pluripotency and post-implantation markers, a slight decrease of the mesoderm gene signature, as well as a concomitant preferential induction of both ectoderm and endoderm markers (Fig. 6d, e). Interestingly, similar gene expression changes were observed at day 4

in all tested conditions, yet the ectoderm and endoderm signatures are expressed at higher levels in si-Rsl24d1 ESCs compared to si-CTL cells (Fig. 6d, e). This result suggests that Rsl24d1 depletion has a moderate impact on lineage commitment choices and/or early dynamics of ESC differentiation.

To evaluate if the differentiation capacity of mesoderm lineages is also affected by RSL24D1 depletion, shRNA-treated cells were assessed by EB differentiation assays[44]. Compared to sh-Control cells, POU5F1 knockdown strongly impaired the capacities of CGR8 ESCs to form viable 12 day-old EBs (Supplemental Fig. 6f) and completely abolished the presence of cardiomyocytes responsible for spontaneous EB beating (Fig. 6f)[44], likely reflecting its preponderant role in early cell fate determination[45,46]. Conversely, RSL24D1 depletion had no significant impact on EB formation compared to sh-Control ESCs (Supplemental Fig. 6f) but partially impaired the differentiation of cardiomyocytes (Fig. 6f). Therefore, decreased Rsl24d1 expression in ESCSs appears to slightly promote ectoderm/endoderm commitment at the expense of mesoderm differentiation. Altogether, these results show that a high RSL24D1 expression in ESCs is required to sustain self-renewal and to support a proper lineage commitment.

## Discussion

Despite displaying reduced translation activity compared to differentiated progenies[14], ESCs paradoxically express RBFs and RPs at higher levels than differentiated cells[21–23]. This observation, therefore, suggests that naive ESCs need to accumulate a pool of ribosomes sufficient to sustain rapid proteome changes necessary to achieve differentiation programs in response to environmental signals and within an appropriate timing. In this study, we highlighted mechanisms coordinating the highly regulated production of ribosomes in ESCs and their contribution to cell fate commitment.

Indeed, we characterized for the first time the molecular function of RSL24D1, a homolog of yeast Rlp24, in the biogenesis of mouse ribosomal LSU. We determined that, in mouse ESCs, RSL24D1 is predominantly localized in nucleoli, which are nuclear domains playing a central role in the early steps of pre-ribosome assembly, and, to a lower extent, in the cytoplasm. Considering the strong conservation of RSL24D1 structure and localization relative to its yeast homolog Rlp24, it is therefore likely that RSL24D1 shuttles between the nucleus and the cytoplasm[32]. In yeast, Rlp24 is assembled at the initial steps of pre-60S maturation[47], together with Tif6, Nog1, and Mak11[16,48], and then allows the recruitment of the hexameric Drg1 AAA-ATPase[32,49]. As the pre-LSU particles exit the nucleus, Drg1 gets activated by nucleoporins and releases Rlp24 by mechanical force, therefore allowing its substitution by the canonical ribosomal protein RPL24[32,49,50]. Unlike Rlp24 deletion in yeast[32], which induced moderate alterations of the 35S and 27SB rRNA intermediates, human cells treated with si-RSL24D1 did not show significant modifications of the rRNA processing[51]. We found that RSL24D1 depletion impaired EIF6 (Tif6 homolog) recruitment to nuclear pre-ribosomes and RPL24 presence in cytoplasmic ribosomes (Fig. 3d). Interestingly, RSL24D1 and EIF6 show different dynamics of disassembly from cytoplasmic ribosomes (Fig. 2c) with EIF6 dissociation occurring belatedly as it is more stably associated with cytoplasmic ribosomal particles. These observations are also in agreement with the yeast model describing Tif6 removal as the latest maturation step of 60S particles[16,52].

RSL24D1 depletion in mouse ESCs caused a decrease of 60S and 80S subunits relative to 40S particles, consistent with observations upon Rlp24 depletion in yeast[32] or 60S biogenesis repression[26,53]. Moreover, we detected the presence of halfmers for di- and tri-ribosomes in si-Rsl24d1 treated cells (Fig. 3b), which were previously observed upon LSU biogenesis alterations in yeast[54,55]. RSL24D1 loss was also correlated to a global reduction in RPL expression while only few RPS were affected (Fig. 3g). Among RPLs, RPL24 was one of the most affected proteins while RPL8 displayed a more moderate

downregulation, thus confirming western blot analyses (Fig. 4d). These observations may result from a loss of nuclear pre-60S and/or of cytoplasmic mature 60S, following the activation of the NDR (non functional rRNA decay) or 60S ribophagy pathways, for example. Further experiments would be required to specify the exact mechanisms responsible for this specific RPL downregulation. Surprisingly, rescuing RSL24D1 expression in ESCs almost completely restored the accumulation of 80S in ESCs while only partially restoring 60S biogenesis (Fig. 3c). This can result from the partial RSL24D1 expression rescue (about 80%), from an efficient recruitment of 60S subunits to 80S in ESCs, or from a difference in turnover rates between free and monosome-associated 60S subunits. Altogether, our data strongly support evolutionarily conserved functions for RSL24D1 to guide RBF association in early steps of LSU assembly, as well as to stabilize ribosome structure locally by protein mimicry until it is replaced by RPL24 during the late cytoplasmic maturation steps.

We established that the impaired accumulation of LSUs in the cytoplasm upon RSL24D1 depletion strongly reduced the global protein synthesis similar to the downregulation of either HTATSF1, which controls rRNA processing, or SSU biogenesis factors in mouse ESCs[19,23]. In addition to a global translational impact, our study highlighted that RSL24D1-dependant ribosome biogenesis tightly controls the steady-state level of specific unstable proteins, including POU5F1 and NANOG, and also of several PRC2 factors, including EZH2, EED, and SUZ12 in ESCs. Although these factors play a key role in organizing the landscape of transcriptionally silenced chromatin by maintaining H3K27me3 modifications, particularly on promoters of repressed developmental genes, their activity rather seems rate-limiting in ESCs. Indeed, the downregulation of RSL24D1 induced a reduction in *Eed*, *Suz12,* and *Ezh2* mRNAs expression accompanied by a reduction of their association to polysomes, likely to affect the neo-synthesis of corresponding proteins, which is correlated to a reduction ranging from 7 to 35% in EED, SUZ12 and EZH2 protein levels. Although EZH2 is responsible for the methyl-transferase activity, previous studies have demonstrated that affecting other core components of PRC2 also led to developmental defects[8], thus suggesting that RSL24D1 depletion may have a more pronounced impact on PRC2 activity than on the expression of its components. Indeed, despite a modest PRC2 protein level decrease, we observed both reduced levels of H3K27me3 in ESCs (Supplemental Fig. S4f) and a significant increase in expression, at the mRNA level, for hundreds of PRC2 target genes (Fig. 4e) which correspond to developmental genes that are normally associated to repressive H3K27me3 marks. RSL24D1 depletion caused the upregulation of over 138 PRC2 target genes associated to the "Development process" GO annotation ($p < 7.3E{-}20$), out of which 120 (87%) display H3K27me3 marks on the vicinity of their promoter in ESCs according to the ChIP-Atlas database. These striking correlations should however be validated by performing ChIP-seq analyses of PRC core factors and H3K27 methylation marks from si-Rsl24d1 treated ESCs to characterize the modifications of their distribution on the chromatin landscape of ESCs and experimentally confirm that the activity of PRC2 is impaired in absence of RLS24D1. Similarly, although analyzing previously published ChIP-seq and target signatures derived from mouse ESCs provided interesting predictions about PTFs target genes likely affected by RSL24D1 loss (Fig. 4c, d), performing combinatorial ChIP-seq analysis on SOX2, NANOG, and POU5F1 in CGR8 cells treated with si-CTL or si-RSL24D1 would be necessary to clarify the direct impact of RSL24D1 on pluripotency transcriptional programs.

At the cellular level, RSL24D1 depletion caused a significant loss of proliferation, likely resulting from ribosomal stress activation by 60S RP loss[36], supported by TP53 and MDM2 upregulation (Fig. 3g). We also observed an increased proportion of ESCs in the G1 phase, suggesting a lengthening of the G1 phase. These observations, correlated to the increased expression of post-implantation and lineage commitment markers, suggest that ESCs are more prone to exit the

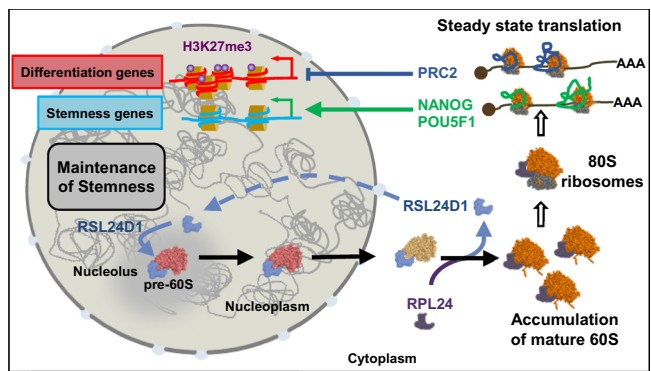

**Fig. 7 | Model describing the functions of RSL24D1 in the LSU biogenesis and in the translational regulation of key pluripotency and developmental programs in mouse ESCs.** Some of the content displayed in this figure were created with BioRender.com, under full licence agreement.

naive pluripotent state upon RSL24D1 downregulation. Accordingly, we observed a decrease of ESC self-renewal capacities (Fig. 6c–e) as well as a stronger propensity to initiate differentiation into endoderm and ectoderm (Fig. 6e) when RSL24D1 is depleted. Surprisingly, RSL24D1 depletion also caused a reduced formation of cardiomyocytes as ESC differentiate in EB structures for 12 days (Fig. 6f), without affecting EB formation (Supplemental Fig. 6f), suggesting that ESCs are also less prone to differentiate into fully functional mesoderm-derived lineages. These observations should further be confirmed by in vivo approaches to assess the pluripotency capacities of RSL24D1-depleted ESCs.

Altogether, this therefore suggests that the versatile status of ESCs, which must rapidly switch from a transcriptionally active self-renewal program to the activation of poised differentiation programs, requires a finely tuned translational activity that is coordinated with ribosome biogenesis in ESCs. We propose a model where RSL24D1 is highly expressed in naive mouse ESCs, shuttling back and forth between the nucleoli and the cytoplasm, to sustain elevated rates of pre-60S biogenesis and steady-state translation (Fig. 7). Conversely, RSL24D1 activity seems less critical in differentiating mouse ESCs, likely reflecting a reduction in cell proliferation during cell fate changes. Additionally, the elevated biogenesis rate in ESCs could allow the accumulation of a stock of ribosomes sufficient to accommodate all the transient translational changes that support cell commitment and differentiation without relying on fluctuating levels of de novo ribosome biogenesis. Once differentiated, cells may require variable levels of ribosome biogenesis depending on their proliferative capacities or metabolism requirements, as seen for MEFs and NIH3T3.

RSL24D1-mediated ribosome biogenesis, therefore, appears to have a dual role in maintaining ESC self-renewal and proliferation (Fig. 7). On the one hand, RSL24D1 maintains a balanced expression of key PTFs, including POU5F1 and NANOG, to control pluripotency transcriptional programs. Furthermore, RSL24D1 is a predicted target gene of POU5F1 and NANOG (Supplementary Data 6 and ChIP-Atlas database), suggesting that RSL24D1 expression could be directly controlled by POU5F1 or NANOG in mouse ESCs. On the other hand, RSL24D1 sustains steady-state translation of PRC2 factors to maintain repressive H3K27me3 chromatin marks and prevent the premature activation of developmental genes in naive ESCs (Fig. 7). Intriguingly, EZH2 was recently shown to enhance 2′-O-methylation of rRNAs by promoting the interaction between NOP56 and FBL in mouse extra-embryonic endoderm stem cells[56]. This observation raises the question of whether RSL24D1 might influence rRNA 2′-O-methylation by modulating EZH2 expression in naive and differentiated ESCs.

RSL24D1 appears to be at the core of a regulatory loop that coordinates ribosome biogenesis and translation with chromatin

epigenetic and transcription regulations. Such regulatory loop has a dual function to support ESC long-term self-renewal while ensuring a rapid gene expression rewiring to orchestrate lineage commitment in response to appropriate differentiation stimuli. Therefore, modulating the production of ribosomes could also be a mechanism to control the translation of specific subsets of mRNAs that regulate ESC fate. Indeed, it is puzzling that despite an active ribosome biogenesis, ESCs display a globally reduced translational activity[14]. One hypothesis could be that controlling the ribosome concentration may regulate substrate selectivity. This "Ribosome Concentration" model has been previously discussed[57] and may explain why the translation of ESC-relevant mRNAs requires a specific amount of ribosomes, and why perturbations of ribosome biogenesis affect specific mRNA populations, such as upstream open reading frame containing mRNAs[58,59]. One could therefore propose that proper levels of ribosome biogenesis in ESCs are required to control the translation of specific gene programs and can be adjusted to promote a timely and efficient proteome rewiring during cell identity changes. Along these lines, ribosome biogenesis defects impacting gene-specific regulations have been proposed to participate to the etiology of congenital disorders called ribosomopathies, which are caused by mutations in RBFs or RPs[57]. Therefore, better defining the molecular feedbacks between ribosome biogenesis, translation, and additional key steps of gene expression, including chromatin modifications and transcription, could not only benefit to developmental biology but also open novel avenues to consider disease treatments.

# Methods

## Cell culture

Mouse CGR8 ESCs (ECACC General Collection, 07032901) were cultured on 0.2% gelatin-coated plates either in ESC$^{FBS}$ conditions using GMEM BHK-21 (Gibco) supplemented with 10% ESC-grade heat-inactivated Fetal Bovine Serum (HI-FBS), 1× non-essential amino acid (NEAA, Gibco), 2 mM sodium pyruvate (Gibco), 100 µM β-mercaptoethanol (BME, Sigma-Aldrich), 10$^3$ U/ml Leukemia Inhibitory Factor (LIF, StemCells Technologies) or in ESC$^{2i}$ conditions using 45% DMEM/F12 (Gibco), 45% Neurobasal medium (Gibco), 100 µM BME, 1.65% Bovine Serum Albumin Fraction V (Gibco), 1× penicillin-streptomycin (P/S, Gibco), 1% N2 supplement (Gibco), 2% B27 supplement (Gibco), 1× ESGRO 2i supplement (Millipore) and 10$^3$ U/ml LIF (Millipore). Mouse ESC lines R1 and G4 were cultured on irradiated MEFs in ESC$^{FBS}$ media. Total cell extracts of H9 and OSCAR human ESCs were provided by the Savatier laboratory[31] (Stem cell and Brain Research Institute, Bron, France). CD-1 MEFs (StemCell technologies) were cultured according to the manufacturer's recommendations. HEK-293T cells (ECACC, 93061524) and NIH-3T3 cells (ATCC, CRL-1658) were cultured in DMEM (Invitrogen) supplemented with 10% HI-FBS, 1× NEAA, 2 mM sodium pyruvate and 1× P/S. All cell lines were confirmed mycoplasma-free (Lonza, MycoAlert kit). CGR8-RESCUE ESCs were generated by transiently transfecting pBASE and PB-TAG-ERP2 (1:8 ratio) and then selecting 48 h with puromycin (1.5 µg/ml). For inducing rescue expressions of RSL24D1, ESCs were treated with doxycycline 100 ng/ml at least 48 h before experimental assays.

## MEF isolation for iPSC reprogramming

MEFs were isolated from R26$^{rtTA}$;Col1a1$^{4F2A}$ E13.5 embryos after removal of the head and internal organs[60]. The remaining tissues were physically dissociated and incubated in trypsin 10 min at 37 °C. Dissociated cells were resuspended in MEF medium (DMEM supplemented with 10% fetal bovine serum, 100 U/mL penicillin/streptomycin, 1 mM sodium pyruvate, 2 mM L-glutamine, 0.1 mM Non-Essential Amino Acids and 0.1 mM β-mercaptoethanol).

## iPSCs reprogramming

R26$^{rtTA}$;Col1a1$^{4F2A}$ MEFs were plated in six-well plates at 10$^5$ cells per well in MEF medium for 24 h and then in MEF medium containing 2 µg/mL doxycyclin[61]. MEFs were reseeded after 72 h on 0.1% gelatin-coated plates in iPSC medium (DMEM containing 15% KnockOut Serum Replacement, 10$^3$ U/mL LIF, 100 U/mL P/S, 1 mM sodium pyruvate, 2 mM L-glutamine, 0.1 mM NEAA and 0.1 mM BME). The medium was replaced everyday with doxycyclin-containing fresh medium for 14 days.

## Plasmids

pLKO.1-puro plasmids (Addgene #8453) were cloned as previously described[62]. Briefly, the pLKO.1 plasmid was digested with EcoRI and AgeI restriction enzymes. Linear plasmids were ligated with annealed oligomers containing the shRNA sequence flanked by EcoRI and AgeI restriction sites. All clones were sequenced to confirm the insertion of each shRNA sequence prior to lentiviral production. shRNA sequences used in this study are reported in Supplementary Data 8A. For rescue experiments, PB-TAG-RSL24D1 plasmid was generated by gateway cloning (Invitrogen). Briefly, mouse ESC Rsl24d1 cDNA (ORF) was PCR-amplified to introduce attb1 and attb2 site in 5' and 3', respectively. Rsl24d1 ORF was subcloned in pDONR221 by BP gateway cloning according to the manufacturer's recommendations. Sequence-validated clones were then cloned in PB-TAG-ERP2 (gifted by K. Woltjen, Addgene plasmid # 80479[61]) by LR gateway cloning (Invitrogen) according to the manufacturer's recommendations to generate the PB-RSL24D1-ERP2 plasmid.

## Lentiviral production and infection

Lentiviral particles were produced as previously described[62]. 3 days after cell transfection using FuGENE HD (Promega), each cell media was concentrated using centricon column (Vivaspin 20, Sartorius) and each viral production was titrated. ESCs were infected with an MOI = 1 for 16 h in ESC$^{FBS}$ conditions supplemented with 8 µg/mL polybrene (Sigma-Aldrich). 24 h after infection, cells were selected with 1.5 µg/mL of puromycin (Sigma-Aldrich) for at least 48 h. shRNA sequences are available in Supplementary Data 9A.

## siRNA transfection

ESC$^{FBS}$ CGR8 cells were transfected with 20 nM siRNAs using DharmaFECT 1 (Horizon Discovery), according to the manufacturer's protocol, and collected at 72 h. siRNA sequences are available in Supplementary Data 9A.

## Colony-formation assay

For sh-mediated depletions, ESCs were plated at clonal density (100 cells per cm2) on gelatin-coated plates in ESC$^{FBS}$ conditions. After 7 days of sh-containing lentivirus infection and puromycin selection (1 µg/ml), culture dishes were imaged and alkaline phosphatase-positive colonies were detected (Alkaline Phosphatase detection kit, Merck-Millipore) and quantified using ImageJ analysis software (Amersham)[63]. For siRNA-depletion, 2000 cells were seeded in 24-well-plates and transfected 24 h with siRNA as described above. Alkaline phosphatase-positive colonies were detected using FRV-Alkaline (Alkaline Phosphatase detection kit, Sigma-Aldrich,), according to the manufacturer's protocol, and quantified using ImageJ analysis software (Amersham).

## Proliferation assays

The proliferation was monitored using the high-definition automated imaging system IncuCyte (Essen BioScience), according to the manufacturer's instructions. 750 cells were seeded in 24-well plates. Proliferation rates were estimated from 2 h interval snapshots according to the manufacturer's instructions.

## Cell Cycle analysis

Cells were transfected with siRNAs in 6-well plates for 48 h, and then incubated with 10 μM EdU (Invitrogen) for 2 h at 37 °C. Incorporated EdU was covalently conjugated to Alexa 647 using the ClickiT® EdU Flow Cytometry Assay Kit (Invitrogen) according to the manufacturer's protocol. In parallel, DNA content was labeled with FxCycle ™ violet stain at 0.6 μg/mL (Invitrogen). Data was collected using LSR Fortessa Cytometer at 647 nm for excitation and a 670/30 bandpass for detection of the EdU Alexa Fluor 647 azide. An excitation at 405 nm and a 450/50 bandpass were used to detect the FxCycleTM Violet (Invitrogen) fluorescence. Data was analyzed using the FlowJo software.

## Differentiation assays

For early differentiation assays, infected cells were cultured after 48 h of selection, at low density ($2 \times 10^5$ cells/10 cm plate) on 0.2% gelatin-coated plates, as previously described, without LIF and with 10 nM retinoic acid (Sigma-Aldrich). The media was renewed every day and cells were collected after 96 h. Embryoid body (EB) formation was performed by the hanging drop method[44]. Briefly 400 cells were cultured in 20 μL hanging drops for 2 days in GMEM BHK-21 supplemented with 20% HI-FBS, 1× NEAA, 2 mM Sodium Pyruvate, 100 μM BME. EBs were then collected in non-adherent culture dishes and cultured for three additional days. At Day 5, cell aggregates were cultured on 0.2% gelatin-coated dishes with 10 nM retinoic acid (Sigma-Aldrich), and the media was changed every 2 days.

## Western Blots

Cells were lysed in 1× Laemmli buffer (50 mM Tris-HCl pH 6.8, 2% SDS, 5% BME, 10% glycerol, and 0.05% bromophenol blue) and protein extracts were analyzed by SDS-polyacrylamide gel electrophoresis (SDS-PAGE) followed by western blotting using antibodies indicated in Supplementary Data 8B. For total protein quantification, 0.5% trichloroethanol (TCE) (Sigma-Aldrich) was included in the SDS-PAGE prior to electrophoresis and activated post-migration with UV light for 45 s. Chemiluminescent and fluorescent signals were acquired on ChemiTouch MP imaging system (Bio-rad) and quantified using ImageLab software (Bio-rad). Serial sample dilutions were loaded onto gels to verify the linearity of quantified signals. Unless otherwise specified, TCE labeling of tryptophan-containing proteins (Total Proteins) was used for normalization of western blot signals. The list of antibodies used in this study is available in Supplementary Data 9B.

## Mass-spectrometry-based quantitative proteomic analysis

$1.2 \times 10^5$ ESC$^{FBS}$ were cultured on 0.2% gelatin-coated 6-well plates and transfected with CTL- or Rsl24d1-siRNAs, in triplicates, as previously described. 48 h after siRNA transfections, the cell pellets were washed with ice-cold PBS and lysed with 200 μl of LYSE-NHS buffer (Preomics), before sonication with Bioruptor Next Gen (Diagenode) at 4 °C and 20 kHz, and heating at 95 °C for 10 min. Protein concentrations were measured using BCA assay (Sigma-Aldrich).

50 μg of each sample were prepared using the iST-NHS kit (Preomics). Peptides resulting from LysC/trypsin digestion were labeled using TMTsixplex Isobaric Label Reagent kit (ThermoFisher Scientific) before mixing equivalent amounts for further processing. The peptide mix was then fractionated using the Pierce High pH Reversed-Phase Peptide Fractionation Kit (ThermoFisher Scientific). Each fraction was analyzed by online nanoliquid chromatography coupled to MS/MS (Ultimate 3000 RSLCnano and Q-Exactive HF, Thermo Fisher Scientific) using a 120 min gradient. For this purpose, the peptides were sampled on a precolumn (300 μm × 5 mm PepMap C18, Thermo Scientific) and separated in a 200 cm μPAC column (PharmaFluidics). The MS and MS/MS data were acquired by Xcalibur (Thermo Fisher Scientific).

Peptides and proteins were identified and quantified using Max-Quant (version 1.6.17.0)[64], the UniProt database (*Mus musculus* taxonomy, 20211122 download) and the frequently observed contaminant database embedded in MaxQuant. Trypsin was chosen as the enzyme and 2 missed cleavages were allowed. Peptide modifications allowed during the search were: $C_6H_{11}NO$ (C, fixed), acetyl (Protein N-ter, variable) and oxidation (M, variable). Minimum peptide length and minimum number of razor peptides were respectively set to seven amino acids and one. Maximum false discovery rates—calculated by employing a reverse database strategy—were set to 0.01 at peptide and protein levels. Statistical analysis of MS-based quantitative proteomic data was performed using the ProStaR software[65]. Proteins identified in the reverse and contaminant databases, and proteins only identified by site were discarded. The abundance of Ubiquitin was separated from those of Rps27a and Fau since these proteins are present as chimeric entries in Uniprot database. After log2 transformation, extracted corrected reporter abundance values were normalized by Variance Stabilizing Normalization (vsn) method. Statistical testing was conducted with limma, whereby differentially expressed proteins were sorted out using a log2(Fold Change) cut-off of 0.3 and a *p*-value cut-off of 0.01, leading to a FDR of to 1.12% according to the Benjamini–Hochberg estimator.

## Immunofluorescence assays

ESC$^{FBS}$ were cultured on 0.2% gelatin-coated coverslip or 96-well plates. 48 h after siRNA transfections, ESCs were fixed 10 min in 4% formaldehyde and permeabilized 10 min in 0.1% Triton X-100. Cells were incubated 1 h in blocking solution (1× PBS, 0.1% tween 20, 5% BSA) and then with primary and secondary antibodies (Supplementary Data 6B). Images were acquired using a Zeiss Axio Imager M2 microscope coupled with the Zen 2 Pro software (Zeiss) and processed with ImageJ.

## High-content analysis system (HCS) analyses

For deeper statistical results, cells were plated on a 96 well carrier plate (Perkin Elmer) optimized for sensitive and resolved fluorescence microscopy. At least 90% of the well surface is acquired in one non-confocal plane with Harmony software on an Operetta CLS Flex High-content-Screening system (Perkin Elmer) equipped with 20×/NA1.0 water objective. The set-up was optimized to reach at least a difference of $10^4$ fluorescence levels between the noise and the signal of interest to allow a robust images analysis and quantification. Data were analyzed with Columbus software (Perkin Elmer). Briefly, on the image of full wells, colonies were located on the nuclei labeling with the appropriate tuned find image region algorithm. To discard small and large colonies, areas in the range of 500 to 40,000 μm² of cells were selected. Then cells in each colony were found with the appropriate tuned find nuclei algorithm. Cell debris and objects with more than one nucleus were excluded by filtering the nuclei according to roundness and surface. Fluorescence intensities are calculated for each selected cellular region for all wells.

## Histone Immuno-histochemistry assays

Cells were centrifugated during 10 min at 377 g and fixed in an alcohol based fixative solution Thinprep® (Hologic) during 15 min. After centrifugated during 5 min at 377 g, the cells were resuspended in 10 ml of Epredia™ Gel (Richard-Allan Scientific™ Histo-Gel™). The gel was hardened during 15 min at 4 °C and the corresponding blocks were dehydrated and embedded in paraffin. 3 μm-sections were immunostained using an antibody anti-histone H3 containing the trimethylated lysine 27 (H3K27me3) (Diagenode). Heat induced antigen retrieval was done using CC1 basic buffer (Ventana). Staining was performed using DAB Ultraview detection system (Ventana).

## Cell fractionation

ESCs were collected and gently lysed for 10 min on ice in hypotonic buffer (HB) containing 10 mM KCl, 0.5 mM MgCl₂, 10 mM Tris-HCL pH 7.4, 1× cOmplete EDTA-free protease inhibitors™ (Roche) and 1 U/μL RNAseOUT (Invitrogen). 0.02% NP-40 was subsequently added for 5 min and nuclei/cytoplasm were then separated by centrifugation. Nuclei were washed with HB supplemented with 0.01% NP-40 and resuspended in Buffer A (250 mM Saccharose, 250 mM KCl, 5 mM MgCl₂, and 50 mM Tris-Hcl pH7.4) with DNAse I (2000 U/mL). Cytoplasmic fractions in HB were adjusted to 250 mM KCL.

## Ribosome purification on sucrose cushion

Cytoplasmic and nuclear fractions were loaded on 1 mL sucrose cushion (1 M saccharose, 250 mM KCl, 5 mM MgCl₂, and 50 mM Tris HCl pH7.4) and centrifuged at 250.000 × $g$ for 2 h at 4 °C. Pellets were washed twice with cold water and resuspended in Buffer C (25 mM KCl, 5 mM MgCl₂ and 50 mM Tris-Hcl pH 7.4).

## Polysome profiling

ESCs were treated with 25 μg/mL of emetine (Sigma-Aldrich) for 15 min and lysed in 10 mM Tris-HCL pH7.5, 5 mM MgCl₂, 100 mM KCl, 1% Triton X-100, 2 mM DTT, 1 U/μL RNAseOUT and 2× cOmplete EDTA-Free protease inhibitors. Lysates were centrifuged at 1300 × $g$ for 10 min to pellet nuclei. Supernatants corresponding to cytoplasmic fractions were then loaded on 10–50% sucrose gradients poured using the Gradient Master (Serlabo Technologies) and centrifuged at 210.000 × $g$ for 155 min at 4 °C. 700 μl fractions were collected using the TELEDYNE ISCO collector while concomitantly acquiring corresponding 254 nm absorbance.

## RNA extraction

**From cells**. Cells were harvested in 1 ml of TRIzol reagent (Invitrogen) and total RNA was extracted according to the manufacturer's instructions. 1 μg of RNA were used for reverse transcription assays using SuperScript II reverse Transcriptase Mix (Invitrogen) according to the manufacturer's instructions.

**From sucrose gradients**. 50 pg of Luciferase RNA (Promega) was added to 250 μl of collected fractions and RNA was extracted with 750 μl of TRIzol LS (Invitrogen) according to the manufacturer's instructions. cDNAs were synthetized using SuperScript™ II Reverse Transcriptase (Invitrogen).

## Real Time qPCRs

Quantitative PCR assays were performed using SYBR Green (Roche, Applied Biosystem) following the manufacturer's instructions. Relative cDNA expression was normalized either by β-Actin, Psmd9, Tbp and 603B20Rik mRNA levels (total cell RNA) or Luciferase mRNAs (sucrose gradient RNA). Serial dilutions were systematically performed to calculate qPCR efficiency, verify amplification linearity and determine normalized relative cDNA concentrations. Primers are listed in Supplementary Data 9C.

## Metabolic labeling of protein synthesis

Cells were transfected with siRNAs in 6-well plates 48 h before metabolic labeling. Cells were then incubated for 5 min at 37 °C with 55 μCi/well of 35S-L-methionine and 35S-L-cysteine Promix (Perkin Elmer). To validate the labeling efficiency, cycloheximide (100 mg/mL final) was added for 10 min prior to labeling. Cells were then washed with 1 ml of ice-cold PBS and lysed in 500 μL of RIPA buffer mixed with 1× final LDS Novex™ 4× Bolt™ loading buffer (ThermoFisher). Extracts were then run on precast Bis-Tris Bolt™ 4 to 12% acrylamide gels (ThermoFisher) before staining with the Simply Blue Safestain (Thermo) according to manufacturer's guidelines to visualize total protein loading across lanes. The gel was then incubated in 30% ethanol, 10% acetic acid, and

5% glycerol for 1 h and dried at 75 °C for 90 min. 35S radioactivity levels were measured using the Typhoon Phosphor imager.

For rescue experiments, cells were transfected with siRNAs in 96-well plates 48 h before HomoPropargylGlycine (HPG) metabolic labeling and treated or not with 100 ng/ml of doxycycline. Cells were then incubated for 30 min at 37 °C with 50 μM HPG in methionine/cysteine-free media. To validate the labeling efficiency, emetine (25 μg/mL final) was added during HPG labeling. Cells were then washed twice with PBS and HPG/Alexa Fluor 594 Click-it reaction was performed according to the manufacturer's protocol (Life Technologies). Alexa Fluor 594 mean fluorescence intensities per cell were determined by HCS microscopy analyses as described above.

## RLP24 homologs protein alignments

The following protein sequences were considered for protein alignments. For RLP24 homologs: *S. cerevisiae* (Q07915), *C. elegans* RLP24 (Q17606), *D. rerio* RLP24 (Q7ZTZ2), *M. musculus* RSL24D1 (Q99L28), *R. norvegicus* RSL24D1 (Q6P6G7), *B. Taurus* RSL24D1 (Q3SZ12), *H. sapiens* RSL24D1 (Q9UHA3). For RL24 homologs: *S. cerevisiae* RL24A (P04449), *S. cerevisae* RL24B (P24000), *M. musculus* RL24 (Q8BP67). Multiple protein alignments were performed with the Clustal Omega software (https://www.ebi.ac.uk/Tools/msa/clustalo/)[66] and visualized with the Jalview software (http://www.jalview.org/)[67].

## RSL24D1 protein structure predictions

Yeast RLP24 and human RSL24D1 structures were obtained from cryo-EM pre-60S structures available in the Protein Data Bank: PDB-6N8J (2019, 3.50 Å resolution)[34], PDB-6LSS (2018, 3.70 Å resolution)[33]. Mouse RSL24D1 protein structures were modeled based on these 2 cryo-EM protein structures using the SWISS-MODEL structures assessment tool (https://swissmodel.expasy.org/assess)[68–70] and were overlapped using the Pymol software.

## RNA-Seq sequencing and analysis

RNA libraries were prepared with the TruSeq Stranded Total-RNA kit and sequenced using an Illumina NovaSeq 6000. Raw sequencing data quality controls were performed with FastQC (0.11.5) and aligned on the mouse genome (GRCm38) with STAR (2.7.0f). RNA quality control metrics were computed using RSeQC (3.0.0)[71]. Gene expression was quantified using Salmon (0.14.1) from raw sequencing reads, using the annotation of protein coding genes from gencode vM20 as index. The analyses were performed using R (3.6.1). Starting from salmon transcript quantification, we used Tximport (v1.12.3)[72] and DESeq2 (v1.24)[73] to perform the differential expression analyses (Wald test, and *p*-value corrections with the Benjamini–Hochberg method). RNA-seq predictions for mouse and human PSCs, differentiated cells and tissues were obtained from a published dataset (GSE45505)[27].

## GO enrichment analysis

The analysis of overrepresented gene ontology (GO) categories were conducted using a reference set of 15093 genes expressed in CGR8 cells and using BiNGO (v3.0.3)[74] as well as the open source bioinformatics software platform Cytoscape (v3.7.2)[75] for visualization of the results. Only annotations with *p* values < 0.01 (defined by hypergeometric test and Benjamini–Hochberg correction) were considered for further analysis.

## Target genes analysis with StemChecker

Differentially expressed genes in Rsl24d1-depleted cells were analyzed using the web-server StemChecker (http://stemchecker.sysbiolab.eu/)[38], without masking the cell proliferation and cell cycle genes.

## Analysis of binding sites defined by ChIP-seq

To identify binding sites for TFs and chromatin-associated factors in the ±1 kb region of TSSs, we combined all datasets available in the

ChIP-Atlas database[39] corresponding to mouse wild type ESCs and mouse differentiated cell types for each factor. Binding sites preferentially bound by selected factors in ESCs were selected if they present a ratio of ESC average score / Differentiated average score >2.

## Reporting summary

Further information on research design is available in the Nature Portfolio Reporting Summary linked to this article.

## Data availability

The publicly available dataset used in this study can be accessed under the GEO accession codes GSE45505, GSE89211, GSE116603, and GSE85717. The RNA-seq data profiles generated in this study have been deposited to NCBI GEO under accession number GSE218290. The mass spectrometry proteomics data have been deposited to the ProteomeXchange Consortium via the PRIDE[76] partner repository with the dataset identifier PXD030497. All other data supporting the findings of the study are available in this article, its supplementary information files, or from the corresponding author upon reasonable request. Source data are provided with this paper.

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

## Acknowledgements

We thank Drs. P. Savatier and P.Y. Bourrillot (SBRI, Bron, France) for providing protein extracts of human ESCs, Dr. M. Zeisel for PRC2 antibodies (CRCL, Lyon, France) and Pr. A. Nagy (University of Toronto, Canada) for providing the R1 and G4 cell lines. We thank R. Pommier, A. Ferrari, and A. Viari from the Gilles Thomas Bioinformatics platform (CRCL, Lyon, France) for the RNAseq analysis, and P. Battiston and T. Andrieu from the flow cytometry platform (CRCL, Lyon, France). We thank L. Belmudes and A.M. Hesse for support in acquisition and analysis of the TMT-based quantitative proteomic dataset. This study was supported by a Marie Curie Career Integration Grant 631794 from the European Union, by an AVENIR grant from the Institut National de la Santé et de la Recherche Médicale (INSERM), by the Ligue Contre le Cancer and by the Institut Convergence Plascan (Grant Number ANR-17-CONV-0002) to M.G., by a Ph.D. fellowship from La Ligue Nationale Contre le Cancer (M.B.), and by grants from Labellisation de la ligue contre le cancer, Agence Nationale pour la Recherche (ANR) and Institut National du Cancer (INCa) (A.H. and F.L.). The proteomic experiments were partially supported by Agence Nationale de la Recherche under projects ProFI (Proteomics French Infrastructure, ANR-10-INBS-08) and GRAL, a program from the Chemistry Biology Health (CBH) Graduate School of University Grenoble Alpes (ANR-17-EURE-0003).

## Author contributions

S.D., M.B., and M.G. designed the research. S.D., M.B., F.B., B.B., J.B., C.I., S.M., A.S., A.H., and D.M. performed the research. S.D., M.B., C.V., F.C., A.A., Y.C., J.J.D., F.L, E.R., F.D., and M.G. analyzed the data. S.D., M.B., F.B., and M.G. wrote the manuscript. All authors discussed the results and commented on the manuscript.

## Competing interests

The authors declare no competing interests.
