## [Peer Review File · Nature Communications]

RSL24D1 sustains steady-state ribosome biogenesis and pluripotency translational programs in embryonic stem cells.REVIEWER COMMENTS

Reviewer #1 (Remarks to the Author):

Translational control is critical to regulate protein expression. Recent studies have shown that ribosome biogenesis and mRNA translation are key pathways that control stem cell homeostasis. However, the underlying mechanisms remain largely elusive. This manuscript started with an examination of over 300 genes involved in ribosome biogenesis and identified RSL24D1 as the most differentially expressed and evolutionarily conserved gene between self-renewing and differentiated ESCs. RSL24D1 is highly enriched in pluripotent stem cells relative to differentiated cells and it functions in the ribosome biogenesis and mRNA translation in ESCs via its association with nuclear pre-ribosomes. The depletion of RSL24D1 impairs global translation, including some key pluripotency factors and PRC2 components, which leads to compromised ESC self-renewal and proliferation, as well as a moderate impact on lineage commitment and cell differentiation.

mRNA translational control in pluripotent stem cells is a relatively underexplored topic, which should be of interest in the related fields. This manuscript has a clear storyline and provides some useful data. However, the manuscript in the current form suffers from a few major weaknesses including the lack of in-depth dissection of the molecular mechanisms, the weak data support for certain conclusions drawn, and the presentation issues (see below). Overall, the manuscript does not meet the standard for publication in Nature Communications. The following points can help authors to improve the manuscript.

Major points:

1. Certain western blots with uneven wells and shrunken bands (e.g., Figure 1B, S1C, 3D A few more) could be further improved for data quality. Particularly for those that rely on signal quantifications to draw conclusions, such a distortion due to the shrunken wells could affect the accuracy of the datasets.
2. Figure 6 and Figure S6 datasets: this section of Rsl24d1 function on lineage choice and differentiation capacity of ESCs should be better characterized with RNA-seq, geneset enrichment, gene ontology, etc. analyses to draw general conclusions, but rather depend on a few cherry-picking lineage markers. The conclusion drawn was not so strongly supported by the minimal marker analyses presented in the manuscript. Some restricting of the main and supplementary figures may be necessary once the new datasets are generated. For example, Fig. S6C could be part of the main figure to support the moderate effect on self-renewal upon Rsl24d1 depletion. Also, it might be better to show the KD phenotype immediately after Fig. 1. Figure S6B western blots should be done for other factors shown in Figure S6A.
3. Data presentation for Figure 4A and S4A: it is unclear why these similar datasets are separated into main figures and supplementary figures. While those related terms were specifically mentioned for the main figure panel, nothing was mentioned for those terms presented in Figure S4A in the results section. So, what is the point of having them and referring to this S4A?
4. The Discussion should be rewritten. The current Discussion is too repetitive with the Results section. It should be presented with better clarity to discuss the major findings and potential caveats or weaknesses of the findings.
5. Fig. S5A-S5C are important datasets and should be presented in the main figures. In Fig. S5C, the authors should add the measurement of Pou5f1 and Nanog mRNA levels in polysome profiling fractions with si-Rsl24d1 and control siRNAs, which should be also in the main figure.

Minor points:

1. All figures: The panels should follow the left to right/top to bottom rules for easy readability.
2. Fig. S1B: Rsl24d1 can be combined with other transcripts in one graph. It would be better to use different colors to distinguish these RNAs.
3. Fig. 3G, 4A, 6B, S3G, and S4A: The authors could use some colors to label the groups to enhance the readability.

4. Page 10 Line 4: "eiF6" should be "eIF6".
5. Page 17 Lines 7-8: "stable long-term depletion", the authors need to tell how long the KD treatment is before the sample collection.

Reviewer #2 (Remarks to the Author):

It has been reported that higher expression of ribosome biogenesis factors and lower translation efficiency are both associated with stem cell function. However, underlying molecular mechanisms that explain these paradoxical observations are not fully understood.

In the manuscript by Durand et al., the authors investigated the roles of RSL24D1, a conserved protein for 60S ribosomal subunit maturation, in mammalian embryonic stem cells (ESCs). The authors first showed that Rsl24d1 mRNA and RSL24D1 protein were highly expressed in mammalian ESCs and induced pluripotent stem cells (iPSCs) compared to differentiated embryonic bodies (EBs) or MEF cells. Having validated the conserved role of RSL24D1 at the late step of 60S ribosomal subunit maturation as reported in its yeast ortholog Rlp24, the authors performed RNAi-mediated knockdown experiments to reveal the role of RSL24D1 in translation and stem cell function. By combining transcriptome and proteomics approaches, the authors demonstrated that RSL24D1 knockdown resulted in perturbation of transcriptional programs mediated by pluripotency transcription factors (PTFs) and polycomb repressive complex 2 (PRC2). This effect was associated with transcriptional and translational downregulation of NANOG and POU5F1 and PRC2 components EED, EZH2, and SUZ12. Finally, the authors showed that RSL24D1 knockdown causes ESC self-renewal and proliferation defects, with a limited impact on the differentiation capacity.

This study reports important findings of RSL24D1 in mammalian 60S ribosome biogenesis and stem cell function. Together with the previous report about 40S ribosome biogenesis (You et al., *Genes & Development* 2015), the current study highlights active ribosome biogenesis as a fundamental characteristic of stem cells. Overall, the manuscript is well-organized, multiple approaches support conclusions, and discussion is appropriate. Several points listed below should be addressed and clarified before publication.

Major points

In Figure 3D, the authors showed that RPL24 amount and its association with pre-60S particles were reduced upon RSL24D1 knockdown. A 60S subunit protein RPL8 was not reduced in this experiment, confirming the specific function of RSL24D1 in RPL24 incorporation to 60S. This result of RPL8 is a bit puzzling, as mature 60S was reduced in Fig. 3B, C. In addition, quantitative mass spectrometry analysis showed that most of the LSU RPs (46/48) were downregulated in RSL24D1-depleted cells (Fig. 3G). Could this be due to the difference in normalization between experiments? Alternatively, is RPL8 one of the proteins observed in the proteomic analysis, which were not affected with RSL24D1 knockdown?

The authors discussed that RSL24D1 supports self-renewal of stem cells, but it is unclear if the proliferation defect upon RSL24D1 is specific to stem cells. Since RSL24D1 is essential in maintaining translation level, RSL24D1 may be more generally required for cell proliferation. The effect of RSL24D1 knockdown on cell proliferation in other non-PSC cultured cells should be tested.

Although the authors performed rigorous experiments using multiple siRNA and shRNAs, no rescue experiments were performed. This reviewer does not think that all data require rescue experiments, but at least some critical experiments should be supported by rescue experiments.

Minor points

In Figure 1F, western blotting detecting RSL24D1 was performed with triplicates. It would be

preferable to perform quantification and statistical analysis.

A significant reduction of LSU RPs in RSL24D1-depleted cells was observed in the proteomic analysis. Is that caused by the destabilization of entire pre-60S ribosomes due to the lack of RPL24 association and subsequent degradation of LSU RPs?

Figures 4C and D compare upregulated and downregulated genes in RSL24D1 knockdown to analyze the frequency of PTFs or PRC2 target genes in each class. Here, it would be appropriate to include a control gene set, such as "all genes" or "unaffected genes," to show enrichment or depletion of target genes.

In Supplemental figure 4B and D, both upregulated and downregulated genes were grouped together. It would be more informative if upregulated and downregulated genes were shown separately.

Could the authors comment on how the ribosome biogenesis is supported in differentiated cells, where RSL24D1 is expressed at low levels?

REVIEWER COMMENTS

We thank both reviewers for their interest and acknowledgment of the work achieved, their careful reading of the manuscript, and their constructive comments which we specifically addressed in the revised version of the manuscript as detailed in a point-by-point response detailed below.

Reviewer #1 (Remarks to the Author):

mRNA translational control in pluripotent stem cells is a relatively underexplored topic, which should be of interest in the related fields. This manuscript has a clear storyline and provides some useful data. However, the manuscript in the current form suffers from a few major weaknesses including the lack of in-depth dissection of the molecular mechanisms, the weak data support for certain conclusions drawn, and the presentation issues (see below). Overall, the manuscript does not meet the standard for publication in Nature Communications. The following points can help authors to improve the manuscript.

Major points:

1. Certain western blots with uneven wells and shrunken bands (e.g., Figure 1B, S1C, 3D A few more) could be further improved for data quality. Particularly for those that rely on signal quantifications to draw conclusions, such a distortion due to the shrunken wells could affect the accuracy of the datasets.

Response: We thank the reviewer for this comment and agree with him/her that quantifications should be based on high quality western blot images as well as reliable quantification methodologies. To this end, we first based all the WB quantifications on total protein detection to ensure non-biased normalization and avoid potential caveat arising from an arbitrary selection of one single control protein. Second, the quantified western blots include serial dilutions of control samples to test for the linearity of the detection and therefore the quantitativity of the assay. Third, all stain-free and immunoblot signals were quantified with the BioRad Image lab software, using the Floating Ball Algorithm for background subtraction (based on Image J rolling ball algorithm), therefore allowing reliable band signal quantifications, which do not depend on band/lane sizes. Fourth, western blot signals were acquired using a ChemiTouch Imaging system and an acquisition mode preventing signal saturation. Finally, the western blots were performed at least in triplicate and quantifications therefore reflects both biological and technical variations.

Thus, in our opinion, the uneven wells and shrunken bands the reviewer is referring to is a consequence of serial dilutions with decreasing amounts of loaded lysates rather than of a distortion of the corresponding gels, or uneven gel qualities. In addition, it is important to recall that these serial dilutions, although important to assess the linearity of our western blots, do not impact the quantification of immunoblot signals, which were only normalized with corresponding stain free blots. We are therefore confident that the quantifications provided in the different figure panels are reliable.

2. Figure 6 and Figure S6 datasets: this section of Rsl24d1 function on lineage choice and

differentiation capacity of ESCs should be better characterized with RNA-seq, geneset enrichment, gene ontology, etc. analyses to draw general conclusions, but rather depend on a few cherry-picking lineage markers. The conclusion drawn was not so strongly supported by the minimal marker analyses presented in the manuscript. Some restricting of the main and supplementary figures may be necessary once the new datasets are generated.

Response: We thank the reviewer for this constructive comment that we have now addressed in the revised version of the manuscript. Following the reviewer's suggestion, we performed a RNA-seq analysis of ESC undergoing a differentiation course, treated or not with siRNAs, and including an RSL24D1 expression rescue to strengthen our conclusions. These assays provide an unbiased analysis of individual gene expression profiles as well as different genes signature based on a set of 60 genes previously published by Semrau and colleagues in 2021 (Nature Communications). Altogether, we provide gene expression data, gene signature scores and Gene Ontology enrichment analyses of upregulated genes between day 0 and Day 4 of retinoic-induced differentiation. As previously described (43), this methodology preferentially promotes ectoderm and endoderm fate over mesoderm, an observation in complete agreement with our results. Therefore, using this assay suggested by reviewer 1, we now show in Figure 6 panels d and e, as well as in a new supplemental figure dedicated to this analysis (Suppl Fig. 7 panels a-c), that ESCs treated with Rsl24d1 siRNAs still commit to ectoderm and endoderm lineages when treated with retinoic acid. Moreover, thanks to these more detailed analyses requested by the reviewer, we've been able to point out that RSL24D1 depletion indeed slightly but significantly promote ectoderm/endoderm differentiation, while decreasing at least partially mesoderm differentiation, a conclusion also confirmed by the decreased presence of cardiomyocytes in EBs without changes in EB formation efficiency. We hope that this new detailed analysis together with the revised conclusions (manuscript pages 19-20) fully address the reviewer's comment.

For example, Fig. S6C could be part of the main figure to support the moderate effect on self-renewal upon Rsl24d1 depletion.

Response: Reviewer 2 requested some rescue experiments to be performed. We therefore propose a revised version of Fig. 6c including these new data for ESC self-renewal, and we kept the shRNA-based data in supplemental Fig. 6 as a confirmation of siRNA-based conclusions (which remain unchanged from the previously submitted manuscript). Following reviewer 1 suggestion, we did integrate the previous panels S6C and 6D for improved clarity into the supplemental figure 6e and hope these new figures fully address the reviewer's comment.

Also, it might be better to show the KD phenotype immediately after Fig. 1.

Response: We thank the reviewer for this comment. We agree that the KD phenotype could also be described earlier in the paper, yet we made the decision to use this data to conclude our study, with a storytelling going from the molecular and biochemical data to the phenotypical data.

Figure S6B western blots should be done for other factors shown in Figure S6A.

Response: As previously discussed, following Reviewer 2's comments, we profoundly revised the figure 6 and supplemental figure 6 with siRNA/rescue experiments and kept some shRNA-based data for validation purposes. Along the same line, we have now provided quantifications of translation efficiency of Oct4 and Nanog mRNAs in si-Rsl24d1 treated ESCs in the current version of supplemental Figure 5b (see below, Reviewer 1 point 5 for more details). Finally, thanks to the reviewer 1's first comment, we now provide RNA-seq based expression data for

a pluripotency signature instead of a few selected genes, including Pou5f1 (Fig. 6d and e). Altogether, we now consider that the shRNA-based data commented by the reviewer 1 are not as robust and detailed as new data added to the revised manuscript and will therefore not provide any additional conclusion to justify their inclusion in the revised manuscript.

3. Data presentation for Figure 4A and S4A: it is unclear why these similar datasets are separated into main figures and supplementary figures. While those related terms were specifically mentioned for the main figure panel, nothing was mentioned for those terms presented in Figure S4A in the results section. So, what is the point of having them and referring to this S4A?

Response: We thank the reviewer for this comment and apologize for the confusion. We have now modified the figure and the text description as requested to improve the clarity of our results.

4. The Discussion should be rewritten. The current Discussion is too repetitive with the Results section. It should be presented with better clarity to discuss the major findings and potential caveats or weaknesses of the findings.

Response: We thank the reviewer for this comment, and we provide a significantly revised and shortened discussion to reduce repetitions, to clarify the message of the manuscript and to discuss some caveats of presented results.

5. Fig. S5A-S5C are important datasets and should be presented in the main figures. In Fig. S5C, the authors should add the measurement of Pou5f1 and Nanog mRNA levels in polysome profiling fractions with si-Rsl24d1 and control siRNAs, which should be also in the main figure.

Response: We thank the reviewer for these constructive comments aimed at strengthening our manuscript. First, as requested by the reviewer, we performed polysome profiling followed by RT-qPCR on Nanog and Pou5f1 mRNAs and added these data in the Supplemental Figure 5 (new panels b and c). These results further support a significant reduction of Nanog mRNA translation upon RSL24D1 depletion. However, the biological and technical variability of this assay on Pou5f1 mRNAs prevented us from drawing a similar conclusion. In addition, these results are consistent with western blot panels that confirms a downregulation of both POU5F1 and NANOG proteins in si-Rsl24d1 treated cells (Supplemental Fig. 5d). Second, we have now reorganized the panels in both Figure 5 and supplemental figure 5, as requested: the figure 5 is focused on polycomb factors while the supplemental figure 5 is more focused on pluripotency transcription factors. We hope this will clarify the message for the readers.

Minor points:

1. All figures: The panels should follow the left to right/top to bottom rules for easy readability.

Response: We agree with the reviewer and modified the panel organization of Figures 1, 2, 4, 5, Suppl. Fig. S1, S3, S4 and S6.

2. Fig. S1B: Rsl24d1 can be combined with other transcripts in one graph. It would be better to use different colors to distinguish these RNAs.

Response: We have modified the panel according to the reviewer's suggestion.

3. Fig. 3G, 4A, 6B, S3G, and S4A: The authors could use some colors to label the groups to enhance the readability.

Response: We thank the reviewer for these comments which were all addressed as followed. For the Figure 3g, we reorganized the figure so that the g panel size could be increased for improved clarity. For the figures 4a (now also including the S4A), S3g (previously S3H), S4c and S7, we added colors to the p-values scales for improved clarity.

4. Page 10 Line 4: “eiF6” should be “eIF6”.

Response: We apologize for this spelling mistake and corrected it accordingly.

5. Page 17 Lines 7-8: “stable long-term depletion”, the authors need to tell how long the KD treatment is before the sample collection.

Response: We thank the reviewer for this point. As described above, we mainly focus the revised manuscript on siRNA-induced phenotypes and RSL24D1 expression rescue experiments. Thus, we have significantly reorganized the corresponding panels in Fig. 6 and Fig. S6 as well as the corresponding text in the result section and we no longer mention the section referred to by the reviewer in the revised manuscript.

Reviewer #2 (Remarks to the Author):

Major points

1. In Figure 3D, the authors showed that RPL24 amount and its association with pre-60S particles were reduced upon RSL24D1 knockdown. A 60S subunit protein RPL8 was not reduced in this experiment, confirming the specific function of RSL24D1 in RPL24 incorporation to 60S. This result of RPL8 is a bit puzzling, as mature 60S was reduced in Fig. 3B, C. In addition, quantitative mass spectrometry analysis showed that most of the LSU RPs (46/48) were downregulated in RSL24D1-depleted cells (Fig. 3G). Could this be due to the difference in normalization between experiments? Alternatively, is RPL8 one of the proteins observed in the proteomic analysis, which were not affected with RSL24D1 knockdown?

Response: We thank the reviewer for this comment and we will now clarify these discrepancies. First, regarding the discrepancies between Fig. 3D and 3B, which compared ribosomes purified either by sucrose gradient or by sucrose cushion: we do indeed agree with the reviewer 2 that this may be due to difference in normalization. Indeed, polysome profiling allows internal normalization (60S decrease is relative to 40S) while this is difficult to internally normalize sucrose cushion data. We intended to use RPL8 as loading control as it is assembled in the early state 2/B (27SB) nucleolar particles, but RPL8 may indeed vary, and therefore, without another robust normalization, we cannot conclude. Thus, we think that polysome profiling is more robust to draw general conclusions. However, it still seems that RPL24 presence in ribosome in Fig 3d decreases more than RPL8 presence in the same purification conditions. Thus, the conclusion that RSL24D1 is required for RPL24 incorporation in 60S is still valid and we have corrected the manuscript to balance our conclusions.

Regarding the discrepancies between Fig. 3d and 3g, which compared total protein from nuclear and cytoplasmic fractions detected by western blot, and total protein extracts quantified by mass spectrometry, respectively. As we did not include serial dilutions to test whether the western blot displayed in Fig3d are quantitative for RPL8, we cannot rule out that variations of RPL8 in total extract exist but cannot be detected in these assays (as a matter of fact, one could consider a slight decrease of RPL8 in nuclear and cytoplasmic extracts in Fig. 3d, compare lanes 1 with 2 and 5 with 6, using HISTONE 3 and GAPDH as loading control, respectively). Therefore, we think that observations in mass spectrometry analyses are more reliable and that

RPL8 expression is decreased upon RSL24D1, similarly to many other RPLs. Accordingly, the mass spectrometry data obtained from total cell lysates showed that while RPL24 belongs to the 5 most affected RPLs in si-RSL24D1 depleted cells (rank 5 out of 46), RPL8 expression is less affected (rank 27 out of 46 with a lower fold change than RPL24).

Overall, these results still support our original conclusions. However, we agree with the reviewer that we should have been more precise, and we have modified the manuscript to clarify that both RPL24 and RPL8 total expression is decreased, yet a variable degree, upon RSL24D1 depletion (mass spectrometry data), but RPL24 presence in ribosome is more affected than RPL8, indicating that RSL24D1 is required for RPL24 recruitment to 60S.

Altogether, these data likely account for the difference of behavior between the RPL8 and RPL24 in the eventuality of deficiencies in later stages of LSU maturation.

2. The authors discussed that RSL24D1 supports self-renewal of stem cells, but it is unclear if the proliferation defect upon RSL24D1 is specific to stem cells. Since RSL24D1 is essential in maintaining translation level, RSL24D1 may be more generally required for cell proliferation. The effect of RSL24D1 knockdown on cell proliferation in other non-PSC cultured cells should be tested.

Response: The reviewer raised here an interesting point. Following reviewer 2's suggestion, we analyzed the proliferation of 2 additional mouse non-PSC cell lines, MEFs and NIH3T3 cells. As shown in the supplemental Figure 6 b,c,d panels, RSL24D1 is expressed at higher levels in mouse ESCs compared to NIH3T3 (about 50% of ESC levels) and MEFs (about 20% of ESC levels) (Fig. 6b). A significant siRNA-mediated depletion of RSL24D1 led to a reduction of proliferation for both MEFs (38% reduction) and NIH3T3 (25% reduction) (Fig S6d), yet the impact of such depletion in ESCs caused a more pronounced loss of proliferation (about 50%, Figure 6a). These results therefore suggest that RSL24D1 is generally required for cell proliferation, in agreement with its constitutive function in ribosome biogenesis described in yeast, and that we confirmed in higher eukaryotes, yet pluripotent stem cells demonstrate an increased dependency on its expression.

3. Although the authors performed rigorous experiments using multiple siRNA and shRNAs, no rescue experiments were performed. This reviewer does not think that all data require rescue experiments, but at least some critical experiments should be supported by rescue experiments.

Response: We thank the reviewer for this interesting proposition to strengthen our manuscript. Accordingly, we generated a stable and doxycycline-inducible rescue model described in the supplemental Figure 3e. With this new model, we now established that RSL24D1 expression rescue is sufficient to partially rescue phenotypes caused by siRsl24d1 treatment, including:

- the loss of accumulation of 60S and 80S ribosomal subunits (Figure 3c);
- the reduction of *de novo* protein synthesis in ESCs (Suppl. Fig. 3f)
- the alteration of gene expression in ESCs (Figure 4b, supplemental figures 4a,b,c,d);
- ESC proliferation (Fig 6a)
- cell cycle deficiency (Figure 6b);
- the formation of pluripotent colonies (Figures 6c,d,e);
- ESC differentiation abilities (supplemental figures 7a,b,c).

All these data are included in the revised manuscript.

Minor points

4. In Figure 1F, western blotting detecting RSL24D1 was performed with triplicates. It would be preferable to perform quantification and statistical analysis.

Response: We have now provided the analysis requested by the reviewer and we also added a control panel for the POU5F1 protein.

5. A significant reduction of LSU RPs in RSL24D1-depleted cells was observed in the proteomic analysis. Is that caused by the destabilization of entire pre-60S ribosomes due to the lack of RPL24 association and subsequent degradation of LSU RPs?

Response: We thank the reviewer for this comment. We agree with the reviewer's conclusion that the mass spectrometry data suggests a global loss of LSU RPs. Because this analysis was conducted on total protein extracts and with a single timepoint, we cannot specifically conclude whether this observation reflects a loss of nuclear pre-60S, a loss of cytoplasmic mature 60S containing RPL24, or an impaired accumulation of pre-60S in the nucleus that cannot be properly exported. The specific alteration of the LSU may result from the activation of a NRD (non-functional rRNA decay) pathway, similar to the yeast 25S NRD (Lafontaine, 2010, Trends in Biochemical Sciences Vol.35 No.5), or from the activation of the 60S ribophagy pathway, for example. What we observed likely results from a combined effect of these different alterations, and defining the causing event(s) of this cascade would be very interesting to pursue in a future study, from a mechanistic point of view. We added this point in the discussion of the revised manuscript (page 22, lanes 503-507).

6. Figures 4C and D compare upregulated and downregulated genes in RSL24D1 knockdown to analyze the frequency of PTFs or PRC2 target genes in each class. Here, it would be appropriate to include a control gene set, such as "all genes" or "unaffected genes," to show enrichment or depletion of target genes.

Response: We thank the reviewer for this comment and modified the figure panels accordingly.

7. In Supplemental figure 4B and D, both upregulated and downregulated genes were grouped together. It would be more informative if upregulated and downregulated genes were shown separately.

Response: We agree with the reviewer and modified the figure panels according to his suggestion.

8. Could the authors comment on how the ribosome biogenesis is supported in differentiated cells, where RSL24D1 is expressed at low levels?

Response: We thank the reviewer for this suggestion and we have decided to include the requested comment as a part of our discussion (Page 24, lanes 565-573)

REVIEWERS' COMMENTS

Reviewer #1 (Remarks to the Author):

The authors have addressed most of my concerns, and the manuscript is greatly improved. However, there still exist a few minor issues.

1. In a few places, the authors wrote "to precise..." the word "precise" was used as a verb, which seems to be an error.
2. Line 372: "Fig. 5d" should be "Fig. S5d".
3. Line 512: "...evolutionary conserved..." should be "evolutionarily conserved".

Reviewer #2 (Remarks to the Author):

In the revised manuscript by Durand et al., the authors performed additional experiments and analyses to address concerns raised by the reviewers. In general, the authors paid a decent effort to improve the manuscript.

Regarding the major points I raised, the authors reasonably explained the discrepancies between western blotting and proteomics data, which is now clarified in the text. The additional cell proliferation analysis with non-ES cells indicated that the cell proliferation defect caused by RSL24D1 knockdown was not specific to stem cells but common in several cultured cells tested (50% reduction of cell proliferation in ESCs vs. 38% in MEFs and 25% in NIH3T3). Although this data does not fully support the authors' original argument, I appreciate that the authors performed this experiment and described the result appropriately. The data now provide important unbiased information to the readers.

Regarding Figure 4d, the authors now include "expressed genes" as a control set in response to my comment. However, the authors still compare upregulated genes with downregulated genes and argue that Pou5f1 and Nanog targets are enriched in downregulated genes. This comparison is misleading as Pou5f1 and Nanog targets are indeed depleted in upregulated genes (~10%) compared to expressed genes (~20%). Downregulated genes are comparable to expressed genes (~20%). This result is inconsistent with the StemChecker analysis in Figure 4c. The description and discussion related to these analyses need to be revised.

REVIEWER COMMENTS

We thank once again both reviewers for their careful reading of the revised manuscript, and their new comments, which have now been addressed in the final version of the manuscript as discussed in the point-by-point response detailed below.

Reviewer #1 (Remarks to the Author):

The authors have addressed most of my concerns, and the manuscript is greatly improved. However, there still exist a few minor issues.

1. In a few places, the authors wrote "to precise..." the word "precise" was used as a verb, which seems to be an error.

Answer: We thank the reviewer for pointing at this inappropriate use of the word “precise” and we apologize for this language distortion. We have now corrected each occurrence in the revised version of the article with “to specify” or “to define”.

2. Line 372: "Fig. 5d" should be "Fig. S5d".

Answer: We thank the reviewer for picking up this error. We have now corrected it in the revised version of the article.

3. Line 512: "...evolutionary conserved...." should be "evolutionarily conserved".

Answer: We thank the reviewer for picking up this mistake. We have now corrected it in the revised version of the article.

Reviewer #2 (Remarks to the Author):

In the revised manuscript by Durand et al., the authors performed additional experiments and analyses to address concerns raised by the reviewers. In general, the authors paid a decent effort to improve the manuscript.

Regarding the major points I raised, the authors reasonably explained the discrepancies between western blotting and proteomics data, which is now clarified in the text. The additional cell proliferation analysis with non-ES cells indicated that the cell proliferation defect caused by RSL24D1 knockdown was not specific to stem cells but common in several cultured cells tested (50% reduction of cell proliferation in ESCs vs. 38% in MEFs and 25% in NIH3T3). Although this data does not fully support the authors’ original argument, I appreciate that the authors performed this experiment and described the result appropriately. The data now provide important unbiased information to the readers.

Regarding Figure 4d, the authors now include “expressed genes” as a control set in response to my comment. However, the authors still compare upregulated genes with downregulated genes and argue that Pou5f1 and Nanog targets are enriched in downregulated genes. This comparison is misleading as Pou5f1 and Nanog targets are indeed depleted in upregulated genes (~10%) compared to expressed genes (~20%). Downregulated genes are comparable to expressed genes (~20%). This result is inconsistent with the StemChecker analysis in Figure 4c. The description and discussion related to these analyses need to be revised.

Answer: We thank the reviewer for his remarks. Regarding his last comment, we believe that inconsistencies raised by the reviewer may rather reflect differences in data sources and processing provided by the stemchecker analysis and the ChIP-seq analysis, yet we agree our interpretation of the data lacked precision. Indeed the Fig. 4d is based on ChIP-seq data and strictly focuses on the +/- 1kb region of gene promoters to detect binding sites (BS) for transcription factors (TF) of interest. This approach is relevant since TF binding sites are often located close to the transcription start site, yet there are also many examples of BS in distant regions, including intergenic regions, which have been proven to control gene expression, so we do not consider our analysis as an exhaustive analysis of functionally relevant binding sites for NANOG and POU5F1. On the other hand, the Stemchecker analysis compares gene lists with curated gene signatures for numerous TF known to be important in different stem cell types. In agreement with the reviewer's remark, we have now modified the corresponding description of the results (lines 315 to 319) and discussion sections to clarify this point (lines 567 to 572) and emphasize limitations of our analyses.